# An Insight into Advances in Developing Nanotechnology Based Therapeutics, Drug Delivery, Diagnostics and Vaccines: Multidimensional Applications in Tuberculosis Disease Management

**DOI:** 10.3390/ph16040581

**Published:** 2023-04-12

**Authors:** Hitesh Chopra, Yugal Kishore Mohanta, Pradipta Ranjan Rauta, Ramzan Ahmed, Saurov Mahanta, Piyush Kumar Mishra, Paramjot Panda, Ali A. Rabaan, Ahmad A. Alshehri, Basim Othman, Mohammed Abdulrahman Alshahrani, Ali S. Alqahtani, Baneen Ali AL Basha, Kuldeep Dhama

**Affiliations:** 1Chitkara College of Pharmacy, Chitkara University, Rajpura 140401, Punjab, India; 2Nanobiotechnology and Translational Knowledge Laboratory, Department of Applied Biology, School of Biological Sciences, University of Science and Technology Meghalaya (USTM), Techno City, 9th Mile, Ri-Bhoi, Baridua 793101, Meghalaya, India; 3School of Biological Sciences, AIPH University, Bhubaneswar 754001, Odisha, India; 4Department of Physics, Faculty of Science, Kasetsart University, Bangkok 10900, Thailand; 5National Institute of Electronics and Information Technology (NIELIT), Guwahati Centre, Guwahati 781008, Assam, India; 6Department of Botany, B. N. College, Dhubri 783324, Assam, India; 7Molecular Diagnostic Laboratory, Johns Hopkins Aramco Healthcare, Dhahran 31311, Saudi Arabia; 8College of Medicine, Alfaisal University, Riyadh 11533, Saudi Arabia; 9Department of Public Health and Nutrition, The University of Haripur, Haripur 22610, Pakistan; 10Department of Clinical Laboratory Sciences, Faculty of Applied Medical Sciences, Najran University, Najran 61441, Saudi Arabia; 11Department of Public Health, Faculty of Applied Medical Sciences, Albaha University, Albaha 65779, Saudi Arabia; 12Department of Medical Laboratory Sciences, Faculty of Applied Medical Sciences, King Khalid University, Abha 61481, Saudi Arabia; 13Laboratory Department, King Fahad Specialist Hospital, Dammam 32253, Saudi Arabia; 14Division of Pathology, ICAR-Indian Veterinary Research Institute, Izatnagar, Bareilly 243122, Uttar Pradesh, India

**Keywords:** tuberculosis, nanotechnology, diagnosis, drug delivery, vaccine

## Abstract

Tuberculosis (TB), one of the deadliest contagious diseases, is a major concern worldwide. Long-term treatment, a high pill burden, limited compliance, and strict administration schedules are all variables that contribute to the development of MDR and XDR tuberculosis patients. The rise of multidrug-resistant strains and a scarcity of anti-TB medications pose a threat to TB control in the future. As a result, a strong and effective system is required to overcome technological limitations and improve the efficacy of therapeutic medications, which is still a huge problem for pharmacological technology. Nanotechnology offers an interesting opportunity for accurate identification of mycobacterial strains and improved medication treatment possibilities for tuberculosis. Nano medicine in tuberculosis is an emerging research field that provides the possibility of efficient medication delivery using nanoparticles and a decrease in drug dosages and adverse effects to boost patient compliance with therapy and recovery. Due to their fascinating characteristics, this strategy is useful in overcoming the abnormalities associated with traditional therapy and leads to some optimization of the therapeutic impact. It also decreases the dosing frequency and eliminates the problem of low compliance. To develop modern diagnosis techniques, upgraded treatment, and possible prevention of tuberculosis, the nanoparticle-based tests have demonstrated considerable advances. The literature search was conducted using Scopus, PubMed, Google Scholar, and Elsevier databases only. This article examines the possibility of employing nanotechnology for TB diagnosis, nanotechnology-based medicine delivery systems, and prevention for the successful elimination of TB illnesses.

## 1. Introduction

Tuberculosis (TB) is a highly infectious bacterial disease that causes illness and death worldwide. A single infectious agent such as TB caused more deaths than HIV/AIDS and COVID-19 combined until the COVID-19 pandemic. Ten million people are infected with tuberculosis each year. Tuberculosis kills 1.5 million people every year, making it the world’s deadliest infectious disease [1,2,3,4]. TB is caused by the bacterium *Mycobacterium tuberculosis*, which is dispersed through the air by individuals who are infected with active TB [5,6]. Patients with lung TB cough, sneeze, and/or spit into the air, which spreads the disease. The disease typically affects the lungs and is called pulmonary tuberculosis, but other regions of the body are also susceptible to the disease. Approximately 90% of people who develop the disease are adults, with men suffering the illness more frequently than women. Even though most people who get TB do so in countries with low and intermediate incomes, TB may be found all around the world. Almost half of the TB-affected people are belonging to southeast countries such as India, Pakistan, Bangladesh, the Philippines, China, Indonesia West African country Nigeria, and South Africa. TB bacteria are believed to be present in around a quarter of the world’s population. One-fourth of the world’s population has suffered from *M. tuberculosis* infection [7]. Almost 5–15% of the estimated 1.7 billion *M. tuberculosis*-infected individuals acquire tuberculosis once in their lifetime; however, immune-compromised individuals are at a higher risk of infection [7].

According to the recent report by the World Health Organization (WHO), there were 5.8 million confirmed TB cases and 1.3 million deaths in 2020, which increased from 1.2 million in 2019 [7]. In contrast, the COVID-19 pandemic has interrupted years of global progress in reducing TB deaths, with the overall death rate reverting to the 2017 level in 2020 and the global TB death rate following the same pattern in the subsequent decades [7]. In 2020, the COVID-19 epidemic had a greater impact on TB mortality than HIV/AIDS. Despite the evidence that tuberculosis is in the decline phase, the establishment and transmission of “multidrug resistance (MDR)” and “extensively drug resistance (XDR)” in *M. tuberculosis* strains has become a serious impediment to TB control globally [8].

The most prominent factors in the origin and transmission of MDR-TB include inadequate management of TB, misuse of antimicrobial drugs, or ineffective drug combinations that trigger the spread to other vulnerable people. The inflation of point mutations in genes encoding drug targets and/or drug-converting enzymes acts as a serious mechanism for gaining resistance to *M. tuberculosis* [9]. There has been an exponential increase in searches related to tuberculosis, as evident from Figure 1. The data has been presented as collected from PubMed, using English as the language with a time interval of 2013–2022 (as of dated 21 March 2023).

## 2. Different forms of TB

*M. tuberculosis* mostly causes lung infection, but it can harm any part of the body. Furthermore, tuberculosis can represent a multitude of diseases, ranging from an asymptomatic infection to a fatal illness [10]. Tuberculosis patients are classified into two groups: latent TB infection, which is asymptomatic and non-transmissible, and those with active TB disease, which is transmissible (in active pulmonary TB) and can be diagnosed by culture or molecular testing. Patients with active tuberculosis have clinical manifestations such as loss of appetite, weariness, fever, and weight loss, while those with pulmonary illness can have a chronic cough with blood and an abnormal chest radiograph, and they are potentially contagious in the advanced disease. While pulmonary tuberculosis is the most common manifestation, the disease can affect almost any organ or cause widespread infection. On the other hand, asymptomatic patients with active, culture-positive illnesses are referred to as having mild to moderate tuberculosis [10].

### 2.1. Extra Pulmonary Tuberculosis (EPTB) 

According to the WHO diagnostic characterization, extra pulmonary tuberculosis (EPTB) is an infection caused by *M. tuberculosis* that affects tissues and organs outside of the pulmonary parenchyma and accounts for 20–25% of total TB cases. The prevalence of EPTB cases has been increasing in recent years [7,11,12,13]. Lymphohematogenous spread of the initial infection and the subsequent latency of the disseminated TB bacteria lead to the development of EPTB. The disseminated TB bacteria may then acquire reactivation in the event of decreased body resistance or increased susceptibility. Any organ might be affected by the disease, which can strike at any stage of life [11,12,14,15,16,17]. It is still unclear why TB bacilli sometimes reactivate in the lungs and other times in other organs [16,17]. Factors influencing reactivation in organs have been linked to female gender, TB contact history, smoking, and end-stage renal illness [18]. There are few studies on unusual EPTB instances [19,20,21,22,23,24,25]. Immunocompromised people, the elderly, undernourished people, prisoners, kids, the homeless, people from low socioeconomic backgrounds, alcoholics, nursing home inmates, those who live in TB endemic areas, and healthcare professionals are among the high-risk populations. Extra pulmonary TB (EPTB) is found in one out of every five cases of TB [26]. Almost every infection should have EPTB on the differential diagnosis list, especially in nations where TB is widespread. Geographical, social, racial, and economic factors affect the ratio of pulmonary TB (PTB) to EPTB [15,16]. The tuberculin skin test, polymerase chain reaction, adenosine deaminase tests, serum interferon-gamma release, and imaging modalities are employed to diagnose EPTB, while biopsies and culture investigations continue to be the gold standard [26]. Most frequently, EPTB affects the lymph nodes (50% of cases), followed by the pleura (18% of cases), the genitourinary system (13% of cases), the bones and joints (6% of cases), the gastrointestinal system (6% of cases), the central nervous system (3% of cases), and the spine (3% of cases) [27]. Depending on the organ it affects, EPTB manifests a variety of clinical and radiological characteristics that frequently resemble those of other diseases [28,29]. EPTB can be difficult to detect, and a biopsy is highly recommended. Diagnosis difficulties lead to treatment being delayed, therapeutic issues arise, expenses rise, and morbidity and mortality rates rise.

### 2.2. Multidrug-Resistant Tuberculosis (MDR TB)

Multidrug-resistant tuberculosis (MDR-TB) results from incomplete or insufficient treatment regimens, is a serious threat to global health, and fuels frequent epidemics that increase morbidity and mortality worldwide [7]. Rapid access to advanced diagnostic tools is required for earlier detection but is limited in many areas. Adherence and tolerability of the pathogen may be difficult to achieve once the MDR-TB diagnosis is initiated. The recent biomedical report revealed that MDR-TB often plays a crucial role in “Post-TB Lung Disease” (PTLD), which brings disability, as well as necessary, rehabilitation to the victims [30]. 

There are different types of drug resistance in TB based on their susceptibility to various antibiotics, such as rifampicin-resistant (RR-TB) or multidrug-resistant (MDR-TB) (resistance to both isoniazid and rifampicin) [31]. In addition to such drug resistance, extensively drug-resistant tuberculosis (XDR-TB) creates a high risk due to its resistance to rifampicin, as well as any fluoroquinolone, and as a last resort, antibiotics such as bedaquiline or linezolid. XRD-TB, as per the WHO, is defined as TB caused by MTB strains that are resistant to any fluoroquinolone and at least one additional group, a drug such as evofloxacin or moxifloxacin, bedaquiline, and linezolid [32]. While pre-XDB-TB can be defined as the TB caused by MTB, which is resistant to any fluoroquinolone, such infections are exceedingly difficult to treat and have negative effects on the recipient who has been diagnosed. Rifampicin resistance was tested in 71% of patients who confirmed pulmonary TB in 2020 while being diagnosed with bacteriological methods, which is an increased percentage worldwide from the previous year’s. Similarly, between 2019 and 2020, the overall number of TB cases reported per year decreased by 18% [7].

With continuous efforts in the development of TB diagnosis, the percentage of persons being treated (MDR/RR-TB) around the world has decreased by 15%. This indicates that almost two out of every three persons who are victimized with MDR/RR-TB each year due to a lack of appropriate access to the necessary treatments [7]. Antimicrobial resistance, as well as medicine unavailability, economic turmoil, and socioeconomic disadvantages in many locations, may have an impact. 

### 2.3. Infection Stages

#### 2.3.1. Primary Stage

Small particles must get past the upper respiratory defenses and settle in the sub pleural portions of the middle or lower lung lobes for infection to develop. Entering the upper respiratory system, droplets with a diameter of less than a few micrometers are less likely to infect the lungs, because the bronchial tubes are more likely to trap them. The most frequent source of infection is due to a single droplet nucleus holding a limited number of organisms. However, to create sickness in those who are more sensitive, just a single organism may be required, but in those who are less susceptible, repeated exposure may be necessary. For the infection to commence, alveolar macrophages must ingest *M. tuberculosis* bacteria [33,34]. Focal pneumonitis is caused by inflammatory cells being attracted to a specific area of the lung due to the presence of bacteria that are not eliminated by macrophages. In the early stages of illness, some infected macrophages migrate to regional lymph nodes (e.g., hilar and mediastinal), from which they may gain access to the circulation and spread infection. Several organs and structures, including the apical-posterior region of the lungs, the epiphysis of long bones, the kidneys, the vertebral bodies, and the meninges, are susceptible to this condition. Moreover, the patients who have acquired partial immunity to TB or environmental mycobacteria due to vaccination or earlier spontaneous infection by *M. tuberculosis* or environmental mycobacteria are less likely to have hematogenous dissemination.

#### 2.3.2. Latent Stage

An infection that is latent is often the result of a primary infection. Approximately three weeks after unrestricted development, the immune system inhibits bacillary replication in 95% of cases before symptoms or indications are observed. It is possible to develop granulomas with necrotic cells and necrotic centers from bacilli foci in the lung or elsewhere. However, as long as the balance between the host’s resistance and microbial virulence continues, tuberculosis may remain latent, become inactive, or reactivate [35,36]. The infectious foci may appear as fibro nodular scars (Simon foci) or small consolidation zones (Ghon foci) in the apices of either lung. If calcified, Ghon complexes with lymph node involvement are Ranke complexes [37]. Latent infection sites were previously assumed to be largely dormant; however, this is not the case at all. If the condition progresses rapidly, it could result in pneumonia (sometimes cavitary), pleural effusion, and an enlargement of the mediastinal or hilar lymph nodes (which, in children, may compress the bronchi). Lymphocytic effusions are common in minor pleural effusions and are usually gone within a few weeks or days. This pattern may be more common in young children and immunosuppressed individuals. Even if the lungs show no signs of involvement, extra pulmonary TB may arise elsewhere. Meningitis is the most feared extra pulmonary symptom of TB lymphadenopathy due to its immense mortality rate, whether in young or the old.

#### 2.3.3. Active Stage

While the patients who require immunosuppression and follow the solid organ transplantation phase face the greatest danger, the immunosuppressive drugs (corticosteroids and TNF inhibitors) are also known to trigger reactivation in certain patients [37]. Reinfection rather than latent reactivation of the disease may lead to active illness in individuals. Patients who are exposed to a significant inoculum of bacilli are more likely to be reinfected in locations where TB is endemic. In low-prevalence locations, reactivation of latent infections is most common. By way of delayed-type hypersensitivity (DTH), tuberculosis attacks the tissues and causes necrosis with a gaseous appearance of lung lesions that are often, but not always, cavitary in immunosuppressed patients. However, the pleural effusion is limited in the case of progressive primary TB, although direct extension or hematogenous dissemination may induce it. Empyema, pneumothorax, or both may develop from the disruption of a large tubercular lesion into the lung space. Prior to the advent of chemotherapy, medically induced pneumothorax treatment might result in deadly consequences such as TB empyema and abrupt acute hemoptysis. As the virulence of the organism and the degree of the host’s resistance vary, TB advances in several ways. People from isolated populations (such as Native Americans) may have a shorter course of disease than Europeans and their descendants due to the lack of generations of selection pressure to insist on natural immunity [38]. A detailed overview of various stages of TB has been depicted in Figure 2.

### 2.4. Pathogenesis and Immunology of TB

In people with active pulmonary or laryngeal TB, the sputum holds many organisms, which disseminate through coughing, singing, or other forceful respiratory movements. Due to the high concentration of microorganisms found inside lung cavitary lesions, those who have them are more susceptible to infection. If droplet nuclei (particles with a diameter of fewer than 5 microns) harboring tuberculosis bacteria are allowed to float about in the air, they have the potential to spread for many hours. As a result, it is tough to resuspend the organisms (e.g., by sweeping the floor or shaking off bed sheets) as respirable particles after they land on the surface [39,40]. To resuscitate *tubercle bacillus*, dust fragments may work, but they are not small enough for the lungs to infect. Fomites (e.g., contaminated surfaces, food, and personal respirators) seem to have little effect on the spread of the disease. Active pulmonary TB patients who have not been treated may be just as contagious. Those who have positive sputum smears and positive cultures for *M. tuberculosis* strains are more contagious than those who do not. Patients with cavitary disease have higher sputum mycobacterial loads, making them more contagious. Aside from that, the surrounding environment has an impact. The risk of transmission rises dramatically when people are exposed to large quantities of tubercle bacteria in crowded, poorly ventilated enclosed spaces. Poverty and institutionalization make vulnerable people even more vulnerable. There is a heightened risk for medical professionals who are often in contact with patients who are undergoing treatment.

Immunologists may focus on how lymphocytes respond to a mycobacterial infection, because this response is important for both immunity and the development of disease. Since there is no cellular response, the normal stopping of pathogen transmission does not happen, even though this response is needed for immunity. Since this is the case, there is a strong link between having HIV and getting tuberculosis (TB). People with HIV who have TB are sick because their CD4 levels are much higher than those that make them more likely to get opportunistic infections [41]. There is a significantly changed clinical state and an altered inflammatory response in AIDS patients when the immunopathologic effects of the Mtb infections are examined. Non-infected TB granulomas do not show typical caseous necrosis but, rather, a granulocytic infiltration and necrosis [42]. In the mouse model where the CD4 molecule is genetically altered, this high inclination toward granulocytic participation is also seen [43]. Since CD4 T cells are such an important part of the acquired immune response, they guard against infection while simultaneously supporting the formation of mononuclear lesions and the caseous necrosis necessary for transmission. As a result of the dual function of acquired cellular responses, a significant cellular immune response may be seen at the site of an unresolved illness.

The velocity with which an acquired cellular response manifests itself is the most critical consideration. For a potentially protective reaction to operate, the environment must be such that it is unable to do so. Dosage affects the host’s ability to prevent bacterial growth in a similar way. There are times when an illness may cause the bacterial load in a person’s body to rise to the point where the immune system cannot effectively fight off the infection. An unrestrained primary lesion may evolve into a metastasis; however, the growth of bacteria is restricted in metastatic lesions. An aspirated and spread secondary lesion is possible when the primary lesion is aspirated and distributed to other parts of the body. This approach stresses the importance of kinetics, environment, and control in determining how rapidly the reaction happens and how it may be managed. It predates our awareness of many components of the learned cellular response.

Even though lymphocytes were obviously found in TB lesions, their role remained unclear in the early stages of tuberculosis research. For anti-tuberculosis immunity in both systemic [44,45] and aerosol [46] challenge paradigms, early mouse research established the importance of T cells. Transfer models and subsequently the use of gene-deficient animals demonstrated that CD4 T cells were the key mediators of anti-tuberculous immunity [46,47]. Gene-deficient mice were also used to highlight the significance of macrophage activation mediated by cytokines in the regulation of bacterial growth [48,49]. By observing that HIV-induced CD4 T-cell depletion left patients vulnerable to TB [41] and that those genetically defective for the cytokine-mediated macrophage activation pathway were similarly susceptible to tuberculosis, these investigations established its applicability to the human state [50].

## 3. TB Diagnosis Method and Challenges with the Existing Methods

The diagnosis of tuberculosis is still made using a clinical history, microbiological testing, and a chest X-ray [51]. Drug-resistant tuberculosis (DR-TB) must always be confirmed with microbiological and/or molecular confirmation, because its clinical and pathological features are substantially identical to those of drug-susceptible TB [52]. Drug-resistant tuberculosis transmits in the same way as drug-susceptible tuberculosis and is no more contagious. However, a delay in diagnosing the drug resistance or extended periods of infectivity may enable further transmission and antibiotic resistance development. 

The emergence of *M. tuberculosis* strains resistant to the most potent therapies has brought TB back as a major worry and problem for global public health after decades in which it was almost universally treated. It is important to keep in mind that, if every patient with tuberculosis has access to timely diagnosis and care, there is a good chance that they will all be cured, even if they are carriers of strains that are incredibly resistant. If these individuals are to be treated successfully, protocolized clinical management is required [52].

Routine culture and drug susceptibility testing (DST) can be carried out using genotypic or phenotypic methods. The results will not be available for at least two to three weeks if liquid media are used and up to four to eight weeks if solid media are used, because phenotypic testing needs to be done on mycobacteria that are in the active growth phase in the culture media. Solid media also have higher contamination rates and require a lot of equipment, media, and a well-trained technician [52]. If a choice needs to be made about the patient’s best course of therapy, this delay can be unjustified. In contrast, molecular testing, which uses genetic amplification methods to find mutations in genes encoding for resistance to anti-TB medications, delivers results in 24–48 h. Due to this, such tests should be carried out on all TB patients whenever they are available. DST’s molecular procedures are much faster than culture-based methods in terms of obtaining results. A few commercially available systems are pretty much completely automated and require minimal expertise. As a result, especially in resource-constrained environments, these systems are becoming increasingly popular for DST [31].

On the other hand, based on the information about anti-TB drug resistance gene mutations, a rapid molecular diagnostic line probe assay (LPA) has been set up to find drug resistance in *M. tuberculosis*. INNO-LiPARifTB (Innogenetics Inc., Ghent, Belgium); GeneXpert System (Xpert MTB/RIF, Cepheid Inc., Sunnyvale, CA, USA); GenoType^®^MTBDRplus (for INH and RIF resistance detection); and GenoType^®^MTBDRsl (for fluoroquinolones, ethambutol, and aminoglycosides) (Hain Life Science, Nehren, Germany) are commercially available versions of LPA that are used to detect mutations in *M. tuberculosis* drug resistance genes simultaneously. The WHO suggested GenoTypeMTBDRplus LPA for the fast diagnosis of MTB drug resistance in 2008 [53]. Based on a meta-analysis study in 2008, the GenoTypeMTBDRplus assay has a sensitivity of 98 and 89 percent for detecting RIF and INH resistance, respectively, and 99 percent specificity for both [54].

According to certain investigations, the sensitivity of “Xpert MTB/RIF” is scarce in patients with paucibacillary infection, restricting its application in smear-negative and extra pulmonary illness. The “Xpert MTB/RIF Ultra” was developed to overcome these limitations with multi-copy amplification targets (IS 6110 & IS 1081), along with higher sensitivity [55]. It was revealed that “Xpert MTB/RIF Ultra” was a high caliber than the “Xpert MTB/RIF”, with improved sensitivity. However, when it came to detecting RIF resistance, both “Xpert MTB/RIF Ultra” and “Xpert MTB/RIF” performed effectively [56]. In addition to these, the loop-mediated isothermal amplification (TB-LAMP) and the lateral flow lipoarabinomannan (LAM) assay are two more WHO-recommended point-of-care diagnostics also available. Although, the point-of-care diagnostic procedures have some advantages, as well as disadvantages, the goal is to have quick diagnostic tests that are economical and need little skill in decentralized settings [57]. However, due to the global diversity of MTB clinical isolates, there is a chance that molecular genotyping approaches will have differential efficacy, which could have diagnostic concerns. Furthermore, the mutations that cause resistance could differ by location, providing insight into the disease epidemiology [58].

The introduction of whole-genome sequencing (WGS) is another new advance in MDR-TB diagnosis. Unlike targeted molecular tests, which only look at a small number of target regions, WGS provides a quick and complete picture of the *M. tuberculosis* genotype and the single nucleotide polymorphisms (SNPs) that are responsible for or linked to resistance. This information can be used to determine drug resistance. WGS is also beneficial for establishing genetic relatedness and analyzing transmission episodes in outbreak circumstances. The mycobacterial interspersed repetitive unit-variable number of tandem repeats (MIRU-VNTR) approach was previously employed in the UK to determine epidemics. This could reveal clusters of isolates, but it lacked the precision of WGS to be positive for related transmissions [59]. WGS gives more information on how isolates are connected to each other, whether the isolates are supposed to be part of a similar transmission chain, and the timing and direction of transmission. In fact, these are more important for public health action.

There is still no “gold standard” test for the diagnosis of drug resistant TB. Molecular testing can reveal mutations that give minimal levels of resistance but are nevertheless clinically important and are not identified by culture-based testing. Furthermore, molecular testing is unable to identify all the variations that are known to provide drug resistance. However, it is not necessary to regularly validate the findings of molecular DST with culture-based DST, even if doctors may choose to do so if the clinical picture supports it. Since more than 90% of rifampicin-resistant isolates are also resistant to isoniazid in most countries, a positive molecular test for rifampicin resistance can be regarded as a diagnostic for MDR-TB [31]. In the future, the widespread rapid development of new technologies that enable early identification of drug-resistant tuberculosis would most likely minimize the period of infectiousness of drug-resistant tuberculosis patients [60]. However, since drug-resistant mutations in *M. tuberculosis* do not affect the bacteria’s capacity to propagate, drug resistant tuberculosis is expected to spread more slowly than if resistance came at a fitness cost [61].

### 3.1. Current Therapy for Treatment of TB

Isoniazid (INH), rifampicin (RIF), pyrazinamide (PZA), and ethambutol (EMB) are the first-line medications used in the initial treatment of tuberculosis patients. There are several advantages to using isoniazid (INH), including its ability to penetrate the cerebrospinal fluid (CSF) and its ability to be highly bactericidal [62]. It is still the most successful and cost-effective TB therapy option. Bacterial resistance has increased due to decades of unregulated usage, particularly in East Asia. Isoniazid has unusual side effects such as anemia and agranulocytosis. Only approximately 1 in 1000 individuals may get clinical (generally reversible) hepatitis after INH exposure, whereas 20% will have asymptomatic, transitory aminotransferase increases. Those over 35 are more likely to be infected, as are alcoholics, pregnant women, and those with chronic liver disease. Anorexia, nausea, vomiting, or jaundice are symptoms of hepatic toxicity and require drug discontinuation and liver function testing. Those with symptoms or a high aminotransferase level are withdrawn from INH (or asymptomatic elevations greater than five times normal). After modest aminotransferase increases and the symptoms subside, patients may be safely challenged for 2–3 days with a half-dose. This dosage may be maintained as long as 50% of patients tolerate it [63,64].

According to a new study, the deficiency of pyridoxine (vitamin B6) induced by the drug INH may cause peripheral neuropathy in lactating or pregnant women, cancer, diabetes, or HIV-affected patients [65]. In healthy children and young adults, pyridoxine doses of 25 to 50 mg per day may help avoid this problem. Reduced phenytoin doses are necessary, because INH slows hepatic phenytoin metabolism [66]. Even disulfiram, an alcoholic medication, might have an adverse effect when used with it. Pregnant women may safely take INH. If taken orally, rifampin (RIF) is bactericidal, quickly works in cells and CSF, is easily absorbed, and has good cell and CSF penetration. To prevent a recurrence later, it also destroys latent bacteria in macrophages and gaseous lesions [67]. 

Rifampin’s side effects cause some uncommon side effects, such as jaundice, thrombocytopenia, fever, and renal failure. Hepatotoxicity is less of a concern with RIF than it is with INH. When taking RIF, it is important to be aware of potential drug interactions. Rifampicin and numerous antiretroviral drugs interact in a complicated way, necessitating the intervention of a pharmacologist with extensive experience in these interactions. Pregnant women may safely use RIF [68]. In the first two months of treatment, PZA cuts the treatment time in half, from six months to two months, and inhibits the development of resistance to RIF. Hepatitis and gastrointestinal discomfort are the most common side effects of PZA [69]. In most cases, hyperuricemia is minor, and gout is extremely rarely caused. As a popular treatment for pregnant women, PZA has yet shown to be safe. Oral ethambutol (EMB) is the most tolerated first-line medication. Optic neuritis (25 mg/kg) is a typical adverse effect seen in individuals with renal impairment and higher dosages. Optic neuritis causes difficulty in distinguishing between blue and green, followed by visual acuity. However, ethambutol is safe during pregnancy and has less resistance than other first-line medicines.

### 3.2. Challenges in the Current Treatment Strategies

Although TB clinical treatment is still difficult, it is improving. Oral or intravenous drugs cause hepatotoxicity, nephrotoxicity, ocular toxicity, ototoxicity, and other adverse effects. The oral administration of TB medicines is suggested to reduce the pharmacokinetic obstacles and enhance the bioavailability, as well as therapeutic index.

Due to time duration issues in conventional pharmaceutical therapies, the patients are less likely to stick to current therapies, which explains the infectious recurrence and the development of MDR- and XDR-tuberculosis [70,71]. MDR-TB is becoming more common in underdeveloped nations and challenging the healthcare industry. 

Although present anti-TB therapies are successful, new short-course regimens with additional drugs are urgently required to tackle the different complications related to drug and target selection, as well as patient commitment [72], and modern approaches such as nanoparticles, liposomes, dendrimers, etc. must be devised to counter the traditional therapeutic difficulties. Increased drug loading, greater bioavailability, longer retention in organs, and uniform release of the pharmaceuticals might all lead to a successful drug regimen. This would reduce the frequency and multiple drug dosages, which would lead to a successful drug regimen]. The site-specific delivery of medicines can reduce or eliminate the possible systemic side effects by favoring localized exposure rather than systemic exposure [72,73,74]. Those systemic adverse effects have been garnering increased attention to targeted drug treatments. A detailed overview of the drawbacks of traditional approaches and benefits of nanotechnology is depicted in Figure 3.

## 4. Nanotechnological Approach for Combating TB

### 4.1. Nanotechnology in the Treatment of Tuberculosis

Nanoparticles have demonstrated effective treatments and encouraging results for TB treatment due to several advantages over conventional therapy [75,76,77,78] (Figure 2). Numerous types of nanocarriers have been effective in drug delivery systems for a wide range of administration methods [79,80,81,82,83,84]. One benefit of antituberculosis drugs based on nanoparticles over free pharmaceuticals is controlled and prolonged drug release [85]. Additionally, it lowers the dosage frequency, as well as tackles the problem of poor compliance. The various nanosystems used in TB therapy are mentioned below (Figure 4).

### 4.2. Various Nanosystems

#### 4.2.1. Dendrimers

Dendrimers are synthetic polymeric nanoparticles with globular shapes, three-dimensional structures, and hyperbranched architectures [86]. The size (< 100 nm), customized shape, and molecular weight are some of the specific characteristics of these molecules that enable them for frequent use in biomedical applications, such as a drug delivery method to increase the permeability of nanoconjugates across different biological barriers, to prolong the release of drugs, and to improve the solubility of hydrophobic compounds. The dendrimer family’s polypropylene imine (PPI) and polyamidoamine (PAMAM) were both often used as drug carriers. Further manipulation of the dendrimer’s surface groups may increase the dendrimer’s characteristics and boost the attachment of the drug/guest molecule. The synthesis, characterization, and commercialization of PAMAM dendrimers came first in drug delivery [87]. During drug action activity, besides the permeability of the particles in the pulmonary route, the disposition of these particles relies on several physicochemical parameters, such as particle size, which is a very important feature that roughly predicts the disposition of the inhaled particles [88]. However, the particles (1–5 μm in diameter) are excellent for in-depth lung distribution through inertial impaction and sedimentation processes [89]. Hence, a unique dendrimer-combined nanoparticle known as “porous nanoparticle-aggregate particles” (PNAPs) is proposed for the transportation of the nanoparticles to the inner lung and alveoli area. The matrix of the microparticulate system may comprise just dendrimer nanoparticles or additional inert pharmacological ingredients, i.e., sugars or phospholipids. Once deposited in the lungs, and with the adhesion of lung lining fluid, the matrix of the PNAPs degrades and rapidly releases the dendrimer nanoparticles, which act on the targets.

Ahmed et al. (2021) [90] prepared rifampicin-encapsulating PEGylated PAMAM dendrimers for the treatment of TB. The dendrimer PEGylation increased the nanoparticles’ EE and DL percent, because PEG chains give an extra surface for interaction. Ahmed et al. discovered that, when the PEGylation degree grew, the DL% reduced, perhaps owing to an increase in surface-positive charges due to steric hindrance. The density of PEG chains surrounding the dendrimer may be increased to promote drug solubilization; however, this creates a dense cloud that acts as a physical barrier to drug diffusion [91]. The amount of rifampicin molecules that are electrostatically bound to each other might be reduced by increasing the degree of surface PEGylation. It is also possible that peripheral repulsion might stretch the dendrimer arms, transforming them from dense-core structures into dense-shell structures, which creates interior cavities and increases the loading capacity [92]. The drug loading potential of dendrimers with high PEG concentrations was shown to be diminished. The bi-phasic release of nanoformulations began with a burst and ended with a plateau. PEGylated dendrimers release slower than native or unmodified formulations and free medicines. The total rifampicin release data in PBS showed a negative correlation with dendrimer PEGylation (except for 85 percent PEG). PAMAM dendrimer cores contain more rifampicin than surface amines [93,94]. Several methods have been used to lessen the dendrimer toxicity by altering surface cationic groups to reduce positive charges. One technique to minimize dendrimer toxicity is to protect positively charged groups through dendrimer surface functionalization with PEG chains [95]. 

Polycationic noncytotoxic dendrimers with piperidinium and pyrrolidinium were synthesized by Miganin et al. (2021) [96] and used as antitubercular agents. The technique is based on the phenotypic screening of a newly constructed phosphorus dendrimer library (generations 0–4) against three bacterial strains: attenuated Mycobacterium TB H37Ra, virulent *M. tuberculosis* H37Rv, and *Mangora bovis* BCG. The most effective polycationic phosphorus dendrimers, 1G0, HCl and 2G0, HCl, are active against all three strains, with minimum inhibitory concentrations (MICs) between 3.12 and 25.0 μg/mL. This collection of polycationic phosphorus dendrimers represents first-in-class medications to treat TB infection, may meet the clinical candidate criteria for this high burden of infectious illness, and may play a role in resolving the constant demand for novel treatments.

Molecular dynamics simulations were used to examine the relationship between the antituberculosis medicine rifampicin (RIF) and a fourth-generation poly (amidoamine) (G4-PAMAM) dendrimer [94]. It has been calculated that the RIF load capacity per G4-PAMAM is roughly 20 RIF at a pH of neutral. A MD simulation of the dendrimer (RIF20-PAMAM) with 20 RIF molecules and the dendrimer (RIF20-PAMAM) was carried out at two distinct pH levels (neutral and acidic). While RIF molecules were promptly and virtually instantaneously evacuated from the solvent bulk in the simulation at low pH, the complex was shown to be substantially more stable in the simulation at a neutral pH. An intriguing switch for drug targeting, because the *Mycobacterium* dwells in acidic areas of the macrophage provided by the high stability of the RIF-PAMAM complex and the fast release of RIF molecules in an acidic medium. According to these studies, there are a variety of dendrimers that may be appropriate for the delivery of RIF and its derivatives in terms of stability and pH-dependent release.

#### 4.2.2. Nanoparticle

Nanoparticles can be defined as particles with a size range of 100 nm and smaller [97,98]. These particles can be synthesized biologically or through chemical processes [99]. Nanoparticle formulations may have unique features that are useful for drug delivery, even though the nanoparticle itself may not have any such capabilities. In contrast to nanoparticles, tiny molecules stand for unbound drugs that are not part of a nanoparticle delivery system [100]. There are many different types of NPs, but they can be broken down into three broad categories: organic (fullerenes, graphene, and carbon nanotubes); inorganic (metal and metal oxide NPs such as silver, gold, iron oxide, zinc oxide, and silica); and inorganic (liposomes, dendrimers, micelles, and liposome-based NPs) [101]. Although research into the use of NPs in drug delivery has been extensive, inorganic NPs have been found to have advantages over their organic counterparts in terms of protection and bioavailability, thanks to their ability to provide targeted drug action, reduced adverse reactions in the organism, and increased drug transport and penetration [102,103,104,105,106].

MDR and XDR strains of *M. tuberculosis* were inhibited in vitro by the physico-chemically (non-green) produced AgNPs at concentrations of less than 1 μg/mL [107]. While THP-1 macrophages were reported to be able to phagocytose AgNPs, the intramacrophage antimycobacterial impact was limited. It was found that metallo-composite nanoparticles (MCN), which included AgNPs, had a comparable impact on intra-macroscopic mycobacteria [108]. The antitubercular action of the AgNPs is modest, but they enhance the antitubercular effect of rifampicin, which is already within the phagolysosomal apparatus, as was revealed by the reduction of *M. tuberculosis* colony-forming units by 68% with rifampicin coadministration. A 13-nm-sized spherical AgNP was not effective against intramacrophage *M. tuberculosis*, while the Zn addition performed the antitubercular activity [109]. MRC-5 lung cell lines were likewise shown to be unaffected by the 5Ag:5ZnO report, which has both intracellular antibacterial activity and no substantial toxic effects. In contrast, Mohanty et al. (2013) [110] found that spherical, biogenic AgNPs (50–100 nm) derived from *Trichoderma* spp. coupled with antimicrobial peptides had a strong anti-TB effect against intra-macrophagic *M. marinum* and *M. smegmatis* at levels of 0.1 and 0.5 ppm. Thus, the hypothesized antibacterial mechanism included the generation of superoxide radicals and activation of macrophages by cytokines rather than high amounts of NO. The combination of NPs with rifampin resulted in an improved antibacterial action against *M. smegmatis* in the same research. The spherical chitosan-coated AgNPs (CS-AgNPs) at a 3 ppm dosage were found to have seven macrophages in both the pre- and post-exposure treatments [111]. The disruption of cell membranes or chemical inactivation of thiol-containing compounds were proposed as antibacterial mechanisms. At bactericidal concentrations, CS-AgNPs were shown to be noncytotoxic to RAW264.7 macrophages as well. Gentamycin was found to enhance the antitubercular action of the drug. In the same research, CS-AgNPs were also found to be effective against microorganisms such as *Staphylococcus aureus*, *Pseudomonas aeruginosa*, and *Salmonella typhi* [111].

AgNPs’ antimicrobial and cytotoxic properties depend heavily on the materials used to cover them and their surface charge. Examples of these antimicrobial activities include chondroitin sulfate-stabilized AgNPs [112] that were effective against *Acinetobacter baumannii* and *Pseudomonas aeruginosa* (including multidrug-resistant strains) and trimethylchitosan-stabilized AgNPs [113] that were successful against *A. baumannii* and other fungi, as well as anti-candida activity [114]. Alginate was also used to stabilize the AgNPs, and glucose was used as a reducing agent in the study, which was conducted by Chen and colleagues [115]. No interfering contaminants or intermediates were formed by this procedure, which was carried out at room temperature in an aqueous environment, and more nanoparticles may interact with the mycobacterial cell wall per unit area, that theoretically have a larger biocidal ability than microparticles and revealed that smaller particles have stronger antimicrobial action than bigger ones, as previously reported [116]]. However, due to their increased surface area to volume ratio, smaller AgNPs with spherical or quasi-spherical geometries are more likely to release silver ions [117]. Agrawal et al. [118] used AgNPs for Mtb treatment, whereas Chen et al. [115] used alginate-capped AgNPs to exhibit antimycobacterial efficacy against diverse pathogenic Mtb, including drug-sensitive and drug-resistant strains.

Although ethambutol (ETH) is beneficial, it has a very short half-life and is extremely toxic. To attain an effective serum ETH concentration, dosages of at least 500 mg must be taken orally. However, for patients, adhering to such high doses is difficult due to the gastrointestinal and liver-related side effects. Bioactivation is required for the use of ETH. The bacterial monooxygenase EthA, which is under the control of the transcriptional repressor EthR, is involved in this process. The limited susceptibility of *M. tuberculosis* to this antibiotic is partly due to the low degree of ETH bioactivation by EthA [119,120]. The antimycobacterial activity was greatly enhanced in a mouse model of tuberculosis when a booster was administered intraperitoneally in conjunction with an oral administration of ETH at the same dose [121].

For *M. tuberculosis*, the lungs are the most common location of infection, thus administering the ETH: booster pair through the pulmonary route might be a viable technique to increase effectiveness. The encapsulation of ETH and its booster BDM41906 in biodegradable polymeric nanoparticles has the potential to eliminate the bottlenecks of ETH’s tendency to crystallize and poor solubility in water [122]. The confocal microscopy of TB-associated macrophages showed that the suggested formulations retained their efficiency following integration into nanoparticles. The “green” β-cyclodextrin (pCD)-based NPs showed the best physicochemical features and were chosen for in vivo research among the evaluated formulations. Microsprayer^®^-administered NP suspension was shown to be safe and resulted in a 3-log reduction in the pulmonary mycobacterial burden after six doses compared to untreated mice.

A wide range of nanoparticles, such as liposomes; solid lipid particles; poly-l-lactide (PLGA); and biomass-derived materials (gelatin, chitosan, and alginates), have been evaluated for their ability to transport TB medicines both in vitro and in vivo [123,124] and have been found to be effective. The advantages of these TB drug delivery nanoparticle systems over free drugs are outweighed by the disadvantages of mesoporous silica nanoparticles (MSNs) [125], which provide better structural and chemical stability, uniformity, inherent nontoxicity, the ability to contain exceptionally high concentrations of various cargo, and most importantly, flexibility in incorporating additional design features. It has been found that macrophages may induce cell death and inflammation when exposed to biodegradable polymer nanoparticles (e.g., PLGA); however, MSNs are nontoxic by nature. As a result of their ultra-high interior surface area (1000 m^2^g^−1^), MSNs have achieved drug loading capabilities of up to 50%, which is many orders of magnitude higher than liposomal nanocarriers. As a further benefit, the MSNs may be made in a range of aspect ratios so that they can be precisely targeted to certain cells and tissues. 

It has been found that encapsulation of INH, streptomycin (STM), and RIF within nanoparticles (250-nm average diameter) increased the intracellular accumulation of the drugs and decreased the MIC values of INH and STM for *M. tuberculosis* by three- to four-fold compared to that of the free drug, according to the findings of Anisimova and colleagues [126]. This is a significant improvement over the free drug usually used for treatment. Using biodegradable poly (butyl cyanoacrylate) nanoparticles, Kisich et al. [127] reduced the MIC for *M. tuberculosis* in macrophages by a factor of ten when compared to free moxifloxacin. In animal models of TB, Qurrat-ul-Ainet al. found improved chemotherapeutic effectiveness after the oral administration of tuberculosis medicines encapsulated in alginate- and chitosan-based biodegradable microparticles (90–100 μm in diameter) [128]. When administered intravenously to mice, RIF ecapsulated in 264-nm-diameter gelatin nanoparticles by Saraogi et al. [129] released the drug over time. INH, RIF, and PZA may be delivered in vivo using wheat germ aglutinin-functionalized poly (lactic-co-glycolic) acid nanoparticles that have been found to have a long-lasting therapeutic impact in animal models [130]. When INH and rifabutin were encapsulated in poly (lactic) acid using the aerosol method, they found that the intramacrophage drug levels were 20 times greater than those reached by oral, intravenous, or intratracheal instillation [131]. Pandey et al. [132] nebulized guinea pigs with *M. tuberculosis* and delivered RIF, INH, and PZA combined into solid lipid particles (1 to 2 m in size) to produce a therapeutic response comparable to that of a daily oral treatment.

*M. tuberculosis* was successfully treated with a pH-gated MSN loaded with INH in an in vitro investigation by Hwang et al. [133] earlier. A novel MSN-based INH delivery system was later created and tested in vitro and in vivo for its ability to combat *M. tuberculosis* [133]. Prodrug nanoparticles based on INH are covalently bound to MSNs through a hydroazone bond in this innovative design [134,135]. Acidic circumstances, such as those seen in macrophages following particle ingestion, allow the unaltered INH to be rebuilt in the endolysosomal compartment of the cells. It was then applied as a PEI–PEG copolymer to the pH-responsive MSNs to increase their dispersion and stability. MSNP was created by Clemens et al. [136] to deliver RIF and INH and demonstrated that each MSNP-TB drug formulation resulted in the death of intracellular *M. tuberculosis* in infected macrophages, proving the technology’s efficacy and demonstrating its potential. Due to RIF’s high hydrophobicity and phase change, the drug is kept in the porous interior of MSNPs under aqueous conditions, where it is more stable. Due to their poor loading capacity, unaltered MSNP cannot be utilized to treat *M. tuberculosis* while being capable of delivering RIF. The addition of a positive charge, such as 10-kDa PEI, to the particle surface increases the particle absorption and enables the release of chemicals from acidifying endosomes into the cytoplasm [137,138]. Since the polymer covering can hold some medicine, it might possibly boost the MSNP’s loading capacity. RIF is a zwitterion with an isoelectric point of 4.8, despite its high hydrophobicity and higher solubility in chloroform than in water. pKa is connected to the deprotonation of the phenolic hydroxyl group at position C-8 at a pH of 1.7, and pKa is related to the gain of a proton by the N-4 piperazine group at a pH of 7.9, both at this compound. This is because, at a neutral pH, RIF binds to positively charged polymers, such as chitosan [139]. Consequently, the larger loading of RIF onto PEI-coated MSNP than on uncoated MSNP may be explained. The binding of RIF to PEI-MSNP includes both hydrophobic and ionic interactions, while the binding of RIF to uncoated MSNP is largely hydrophobic. For in vivo delivery, the use of PEI-coated NP-RIF would have two benefits over uncoated NP-RIF: greater binding and loading of RIF and improved absorption by macrophages. NP-RIF incubation with *M. tuberculosis*-infected macrophages for three days had no effect on the absorption rate of drug-loaded MSNP in our investigations. This advantage of PEI-NP-RIF vs. NP-RIF may be even more evident in in vivo administration, because the rate of MSNP absorption by macrophages may have a bigger influence on the efficiency of the delivery platform. 

The nanovalves were made to only open in acidic settings, such as cells’ acidifying endosomal/lysosomal compartments, and to stay closed at neutral pH. By attaching molecular threads to the pores and then adding large β-cyclodextrin molecules that bind to the threads at a neutral pH, pH-operated nanovalves were created. The protonation of the molecular threads causes the β-cyclodextrin-capping molecules to have a lower affinity for the threads, allowing the nanovalves to open and release drug molecules from the MSNP [140]. The pH-gated NP-INH was shown to release the drug from the MSNP under pH control in this study using a fluorometric test and a bioassay. These results showed that the drug is released from the MSNP under pH control. 

Chemical and physical approaches may also be used to synthesize AgNPs. Sodium citrate, sodium borohydride, and dimethylformamide are common chemical compounds used to convert Ag^+^ into metallic silver [99]. Organic NPs, on the other hand, have been successfully produced using environmentally acceptable, pure procedures based on the utilization of apiin, geraniol, and gum kondagogu as reducing agents [141]. Researchers found that synthetically manufactured AgNPs inhibited bacteria better [142], although antimycobacterial activity was also found in AgNPs made from medicinal plant-reducing compounds [143]. Even though we do not yet know how each metal works, both studies indicated that the antimycobacterial activity of AgNPs is greater than that of AuNPs and lower than that of bimetallic NPs (Au–Ag). Using biogenic AgNPs in conjunction with antimicrobial cationic peptides is another potential antimycobacterial method being researched [110].

As with any antibiotic, there is the possibility of resistance developing to certain NPs. There were resistant populations of *M. smegmatis* to AgNPs and AgNO_3_ after exposure to AgNPs, as well as antibiotic cross-resistance [144]. However, resistance to other inorganic compounds such as CuSO_4_ and ZnSO_4_ was not detected. Effluent pumps and other resistance mechanisms may be influencing the resistance to these materials, which must be carefully handled. In addition to suppressing the innate responses of monocyte-derived macrophages in response to *M. tuberculosis* infection, AgNPs have been shown to exhibit antibacterial activities in vitro [145]. To properly comprehend the influence of NPs on in vivo models of infection, it is necessary to take into consideration their potential to alter pathogen-induced immune responses. For *M. tuberculosis*’ metabolism and development, a metal called gallium (Ga) is critical. It is chemically identical to iron (Fe). The active site of enzymes may be rendered nonfunctional by substituting Ga for Fe, and this might be a new approach to antituberculosis treatment. The antibacterial activity of gallium nitrate against *M. tuberculosis* colony-forming units (CFUs) in the lungs, liver, and spleen of mice has been shown in murine TB models. A single pharmacological loading of gallium NPs prevented *M. tuberculosis* development in macrophages for up to 15 days [146]. Even though gold has previously been less effective against mycobacteria than silver, thiol-conjugated AuNPs have shown excellent activity against *M. smegmatis* [147].

A novel form of magnetic nanocomposite with a variety of characteristics might be used to diagnose and treat TB infections by covering FeNPs with a polysaccharide (chitosan) and then loading them with an antibiotic such as streptomycin [148]. Similar to PAA-coated iron oxide NPs, these NPs work along with rifampicin, isoniazid, and norfloxacin to prevent drug efflux in *M. smegmatis* [149]. When employing stabilized, coated NPs, this only occurs when the PAA or iron NPs are used alone. This suggests that metallic-coated NPs might be used as efflux pump inhibitors in innovative ways.

Particulate vehicles have gotten a lot of interest because of their potential to provide medications with the respiratory and internalization qualities needed for powder inhalation treatment. Most of the lipids have been shown to be safe and free of toxicity when administered via the airways [150,151]. As a result of this, dry powder inhaler devices may be used to administer medications to treat pulmonary tuberculosis, allowing them to be deposited on alveolar epithelial cells and transported into the AM. Phagocytosis is the known method for AM entry into macrophages infected with Mtb, which provides effective intracellular drug delivery after only one hour [152,153]. In addition, the internal biodegradation of the lipid matrix might fairly be predicted to provide drug biological activity within infected AM [154]. Prior to this, based on AM passive targeting, Maretti et al. [154] developed respirable solid lipid nanoparticle assemblies (SLNas) loaded with rifampicin (RIF), a therapeutically effective first-line anti-TB medication. After that, mannosylated SLNas were used to make an AM active that targets the macrophage membrane’s mannose receptors (MR), which are overexpressed in Mtb-infected cells [155]. 

LDC-NPs are another feasible alternative to colloidal carriers (i.e., polymeric nano- and microparticles, liposomes, and nanoemulsions) and, having a strong hold in controlled targeting and release of drug, have higher intestinal permeability, as well as greater bioavailability. It has been observed that pharmaceuticals integrated into lipids and stabilized with different surfactants exhibit a high degree of permeability, as both the lipids and surfactants work as excellent permeation enhancers for medications in the gastrointestinal system [156]. The surfactant mixtures are employed to stabilize the LDC-NPs and contribute to the drug’s intestinal permeability by blocking the efflux (P-glycoprotein) pump [157,158]. Finally, LDC-NPs provide the scalability and simplicity of manufacturing from a translational standpoint. In vitro drug release tests on improved LDC-NPs revealed a biphasic release pattern with an early phase of rapid release followed by sustained release over 72 h, and LDC-NPs matched the Higuchi model of drug release kinetics. These findings are promising, since they demonstrate the durability of the synthesis scheme and formulation, which have been confirmed using a variety of analytical techniques. 

The strong reason to pursue the nanoparticle strategy against *M. tuberculosis* is that bacteria are mostly intracellular macrophage inhabitants. Additionally, they are the cells that are most effective in phagocytizing a variety of particles from the blood circulation, provided they are larger than roughly 0.2 μm in diameter [159]. As a result, when nanoparticles (NPs) or microparticles (MPs) (>1000 nm), including antibiotics, gain access to the circulatory system of *M. tuberculosis*-infected animals, the particles may be effectively taken up by *M. tuberculosis*-infected macrophages. There, the polymer is destroyed, enabling the medications to be released locally and directly into the diseased cell. Numerous groups have reported compelling evidence using various kinds of polymeric nanoparticles and microparticles encapsulating various medications against mycobacteria in a variety of different animal TB model systems. Most of these experiments used the polymer poly (lactic-co-glycolic) acid (PLGA), with a few using chitosan or alginate [160,161]. Moreover, the most surprising findings in the realm of tuberculosis are statistics directly comparing traditionally delivered anti-*M. tuberculosis* medications (through oral or lung inhalation) to drugs encapsulated in PLGA. In comparison, with NP-based therapy, antibiotics persisted in the blood for more than a week at quantities greater than the minimum inhibitory concentration required to kill *M. tuberculosis* [162]. Various other researchers’ recommendations for the use of nanoparticles are cited in Table 1.

There have been few reports on the toxicity of zinc oxide nanoparticles on the *M. tuberculosis* system. Nevertheless, there are few data on the harmful effects of alginate nanoparticles with bioadhesive properties on intestinal mucosa treatment, which lengthens the duration of their absorption [174]. Nanoparticles of chitosan, rifampicin, and polyethylene glycol are used in TB therapy-controlled delivery systems [175]. Banu and Rathod [176] employed biogenic silver nanoparticles to prevent *M. tuberculosis* growth. Patil et al. [177] synthesized zinc nanoparticles using *Limonia acidissima* leaf for the treatment of tuberculosis. After the incubation time, zinc nanoparticles exhibited a blue tint in the well, which indicates no bacterial growth, whereas a pink color indicates growth. Between 12.5 and 100 μg/mL of zinc oxide nanoparticles, bacterial growth was suppressed. Nevertheless, at a minimum inhibitory concentration of 12.5 μg/mL, zinc oxide nanoparticles had a moderate impact on bacterial growth. Mycobacteria were sensitive to 100 μg/mL of leaf extract but resistant to all doses of zinc nitrate solution. The minimum inhibitory concentration of standard antibiotics such as pyrazinamide and ciprofloxacin is 3.12 μg/mL, while streptomycin’s is 6.25 μg/mL. The solution of zinc oxide nanoparticles hindered the growth of bacteria by initiating a lipid peroxidation process that caused DNA damage, glutathione depletion, modification of membrane shape, and disruption of the electron transport chain [178]. Concentration and size (12–53 nm) are two crucial parameters that may influence apoptosis in cells. Afterwards, nanoparticles enter the cytoplasm of mycobacterium via endocytosis, and the smaller-sized nanoparticles (12–53 nm) penetrate the bacterial cell membrane, inactivating the enzymes necessary for adenosine triphosphate production [179,180], which leads to the formation of reactive oxygen species and, ultimately, bacterial cell apoptosis [110]. The zinc oxide nanoparticles may react with sulfur or phosphorus-containing soft bases, such as R-S-R, R-SH, RS-, or PR3, hence promoting the loss of a DNA replication capacity [181]. Consequently, sulfur-containing proteins in the membrane or inside the cell and phosphorus-containing components such as DNA are expected to be the nanoparticles’ preferred targets. The inhibition reduces the number of places where mycolic acid is transferred to the cell wall.

Abdel-Aziz et al. [164] green-synthesized chitosan/silver nanoparticles (Ag) using N, N, N-trimethyl chitosan chloride (TMC). The MIC for the TMC/Ag nanocomposite to inhibit mycobacteria was determined to be 1.95 μg/mL. Observing the antimycobacterial activity of the TMC/Ag nanocomposite against *M. tuberculosis* cells during the incubation period, the transmission electron micrograph of a treated *M. tuberculosis* cell revealed TMC/Ag nanocomposite depositions or precipitations on the cell wall after 1 day of nanocomposite treatment. These appeared as electron-dense particulates or granules. Periasamy [182] demonstrated that Ag NPs may physically interact with the cell surface of certain bacteria, resulting in structural alterations and damage to the cell membrane that renders the bacterium more permeable [183]. Several investigations have shown that Ag NPs may stick to plasma membranes and alter their permeability [184,185]. It was followed by cytoplasmic leakage from *M. tuberculosis* cells after 3 days of nanocomposite treatment. After 5 days of nanocomposite treatment, the breakdown of intracellular components led to the creation of empty space inside the cell. At the conclusion of the 7-day incubation period, cytoplasm-free or almost cytoplasm-free lysed *M. tuberculosis* cells were seen at low and high magnification [164]. 

Banerjee et al. [186] studied the effect of lipid nanoparticle formulations encapsulating antituberculosis molecules. LNFs of various size ranges comprised of a spherical lipid bilayer are preferable as nanocarriers to transport drug molecules or antigens, because they can protect their loaded materials from the harsh corrosive surroundings until regulated releases at the specified target areas. This extracellular material is taken up by the early and late endocytic pathways, completing the maturation process via the lysosome gateway, while intracellular material is transported to the lysosome.

#### 4.2.3. Nanoemulsion

A colloidal particle in the submicron range, called a nanoemulsion, may be used to carry drug molecules. In terms of diameter, they vary from 10 nm to 1000 nm. Basically, it is an inorganic, lipophilic, and negatively charged sphere. Emulsions, which are two-phase mixtures, are made up of droplets ranging in size from 0.1 to 100 nm. A thermodynamically unstable solution may be stabilized with an emulsifying agent (emulgent or emulsifier). Emulsifying agents facilitate dispersion by acting as intermediaries or interphases. The term “nanoemulsion” may refer to a “mini-emulsion”, as well as a fine oil/water or oil/water dispersion stabilized by an interfacial layer of surfactant molecules. Due to their small size, nanoemulsions are transparent. A transdermal nanoemulsion gel containing rifampicin was created by Hussain and colleagues. The nanoemulsion gel consisted of Transcutol-HP, labrasol, and capmul PG8 [187]. The optimized nanoemulsion gel (OCNE-1T gel) containing Transcutol revealed a 7.43-, 1.44-, and 1.2-fold greater flux value than the drug solution (DS), OCNE-1, and OCNE-1 gel, respectively, and demonstrated that all of the produced nanocarriers could give therapeutic effects when administered transdermally, but DS could not (7.16 ± 0.12 μg/h cm^2^). A short lag time, a high penetration coefficient, and an increased enhancement ratio revealed that the creation of a gel and addition of Transcutol boosted the permeability [188]. The findings demonstrated that the permeation enhancer and gel carrier improved the penetration of RIF over rat skin owing to an increased residence duration and transient disturbance of the subcutaneous architecture [189]. 

Encapsulation of hydrophilic pharmaceuticals in hydrophobic matrices, such as lipophilic polymers or lipids, and the subsequent nanoscale formulation of the drug-encapsulated matrix as colloidal or stably dispersed nanoparticles are two strategies for increasing the permeability of hydrophilic medications. The “lipid nanoparticle formulations” (LNFs), solid lipid nanoparticles, and nanostructured lipid carriers have gained substantial attention owing to their capacity to overcome numerous colloidal carriers’ limitations [190,191,192]. However, in contrast to lipophilic medicines, it has been shown that these LNFs are incapable of incorporating very hydrophilic pharmaceuticals in a stable and efficient way [193,194]. Although efforts have been made to address this issue in tuberculosis therapy, encapsulating hydrophilic medicines inside the hydrophobic core of the majority of LNFs remains a significant obstacle [195]. Another technique that may be more sophisticated is the design and production of lipid–drug conjugate nanoparticles (LDC-NPs). By covalently linking a hydrophilic drug to lipid components, a water-insoluble lipid–drug conjugate (LDC) is formed, resulting in an amphiphilic prodrug molecule. An appropriate terminal functional group (amino or hydroxyl group) should preferably be present on the chosen drug moiety. This would facilitate conjugation with the reactive species present in the lipid component (carboxyl group) for efficient delivery to the target site.

Nanoemulsion-based nanocapsules have a lipidic core that is particularly well suited for encapsulating a highly hydrophobic chemical such as bedaquiline. As a result, Matteis et al. [196] produced two bedaquiline-loaded nanocarriers consisting of a polymeric shell around a liquid or solid lipidic core containing the medication. The results were encouraging for the in vitro antimicrobial activity of the drug against tuberculosis strain H37Rv and almost unchanged after encapsulation in all the nanocarriers, indicating that the drug was not damaged during the encapsulation process and that it is still as effective as the free drug after encapsulation. Furthermore, bedaquiline-loaded nanocarriers had no cytotoxic impact on the A549, HepG2, and THP-1 cell lines when used at the concentrations required to kill the bacteria in the experiments [197]. 

#### 4.2.4. Liposomes

Liposomes are another way to make it easier to administer anti-TB medications to the patient. Given the extreme selectivity offered by liposomes, they are said to have an “improved permeability and retention impact”. Using both passive and active delivery strategies, liposomes may be directed to organs or tissues. This sort of targeting relates to the transport of liposomes throughout the body in accordance with their usual distribution pattern [198]. Liposomes, used in such drug targeting, are either solely composed of phospholipids or composed of phospholipids and sterols. As a result, passive targeting in pulmonary tuberculosis has a substantial interest when it is used in conjunction with an inhalatory route, because the particles are of a size that allows them to pass through the alveoli and reach the AMs, where they are phagocytosed (if the diameter is small, <5 microns). This passive targeting is made feasible by the innate proclivity of macrophages to absorb particles, which makes it possible to target them passively. However, passive targeting in extrapulmonary tuberculosis is of immense interest when paired with the intravenous and inhalatory methods. Upon entering the bloodstream, the liposomes are readily picked up by phagocytic cells of the MPS, which can gain access to the lysosomes therein. In active targeting, the usual distribution patterns of liposomes are altered because of changes in their structure and content [198]. To deliver any drug to pathologic sites or to cross the biological barriers, the charged lipids are used in conjunction with a ligand (i.e., proteins, peptides, polysaccharides, glycolipids, glycoproteins, and monoclonal antibodies) to achieve the target.

It is somehow quite impossible that liposomes cannot be used in the treatment of poverty-related diseases such as tuberculosis (TB), where many patients cannot afford the lengthy regimen because of the high cost of the synthetic or purified natural phospholipids used in their formulation [199,200]. For the delivery of food ingredients, the use of low-cost, naturally occurring phospholipids to create liposomes was recently proven in a review by Adriana et al. [201]. Yokota et al. [200] showed that crude soybean lecithin may be used to make food liposomes. The crude soybean lecithin is used as an alternative to the more costly synthetic or refined lipids in the INH liposome encapsulation method [202]. To improve the bioavailability of anti-TB medications at the location of interest while lowering their adverse effects and dose frequency, these aspects may be advantageous (through targeted, sustained, and controlled release). The current results imply that crude soybean lecithin may be used to effectively encapsulate medicines in liposomes.

A dry phospholipid formulation known as a proliposome was first investigated by Payne and his colleagues in 1986 [203] as a more stable liposome alternative. As a delivery vehicle, proliposomes’ dry shape makes them more convenient to carry. Among the components of proliposomes are hydrophilic carriers that are coated with cholesterol and phospholipids. Both hydrophilic and lipophilic compounds may be successfully encapsulated through their use. Liquid phospholipid formulations that may make liposomes when mixed with an aqueous media were added to the concept of proliposomes later that year in 1991. Particulate-based proliposomes and solvent-based proliposomes are two distinct forms of proliposomes. The formulation of particulate-based proliposomes relies heavily on the choice of an appropriate carrier. The porosity and ability to store phospholipids on the carrier’s surface are the primary factors in carrier selection [204,205]. Solvent-based proliposomes, on the other hand, are created by dissolving lipids in an organic solvent that is also water-miscible. High phospholipid concentrations in organic solvents are combined with dispersion in water to generate liposomes, which may be extracted.

Through spray-drying soy phosphatidylcholine (HSPC) and cholesterol [206], inhalable rifapentine-loaded proliposomes have been developed for the treatment of pulmonary tuberculosis. Developed proliposomes had a mean particle size of 578 nm, an entrapment efficiency of 72.08 percent, and a zeta potential of 29.40 mV. Rifapentine proliposomes that were spray-dried and spherical had good flow characteristics. Rotahaler^®^ aerosolized rifapentine-loaded proliposomes with FPF and MMAD of 92.50% and 2.62 m, respectively, at a flow rate of 60 L/min in an in vitro aerosolization study. A 24-h controlled release trial with rifapentine proliposomes in PBS revealed a release rate of around 90% (pH 7.4). Rifapentine proliposomes also showed a 13.99-, 6.43-, 6.40-, and 2.35-fold improvement in AUC, AUC (0–24), MRT, and Cmax, respectively, when compared to the drug alone at the same dosage (250 g) in an in vivo pulmonary investigation. Inhalable isoniazid (INH) proliposomes were prepared by spray-drying soybean phosphatidylcholine and cholesterol (1:1) with mannitol as an inert carrier [207]. At a flow rate of 60 L/min, irregularly shaped proliposomes showed fine particle fractions (FPF) and a mass median aerodynamic diameter (MMAD) of 35 percent and 2.99 μm, respectively. This in vitro toxicity investigation found that, in MTT assays, INH-proliposomes did not affect the proliferation of NHBE and small airway epithelial cells, which are linked with the lungs (SAEC). Another study found that INH-proliposomes had no effect on alveolar macrophage activation (AMs) in terms of the production of inflammatory mediators such as interleukin-1β (IL-1β), tumor necrosis factor-α (TNF-α), and nitric oxide (NO). Among other things, INH-proliposomes were more effective against *M. bovis*-infected AM than INH alone in antimycobacterial activity.

A new method of inhaling liposomal dry powder has been developed to deliver biologically active licorice extract (LE) [208] with powerful antitubercular action straight to the lungs. Using the thin film hydration process and freeze-drying, liposomes were turned into a dry powder for inhalation. With a mean aerodynamic diameter of less than 5 nm and geometric standard deviation of around 1.2 nm, the liposomal dry powder for inhalation showed a good in vitro lung deposition profile, as proven by the Anderson Cascade Impactor (ACI). Research into the liposomal dry powder for inhalation’s (LDPI’s) in vivo lung deposition in Swiss albino mice confirmed its in vitro pulmonary drug deposition results. It is estimated that 46 percent of the given medicine reached the lungs, and roughly 16 percent of the given medication was retained in the lungs after 24 h of administration. Antitubercular LE may be sustained and maintained at therapeutically relevant quantities in lung tissue for a long period of time, as shown by this extended retention. Infected animals’ lung and spleen bacterial levels were statistically significantly reduced, as shown by an in vivo pharmacodynamic assessment of the formulation using a mouse experimental TB model. Licorice extract might be utilized to produce a new, safe, and effective TB medication based on this study’s findings. 

#### 4.2.5. Miscellaneous

Sangboonruang et al., {209] fabricated a novel niosome-based platform consisting of an ethanolic extract of propolis (EEP) via the film hydration method [209]. The surface of noisome was modified using the Ag85A aptamer that helps in targeting the Mtb. Ag85 proteins are maintained in the Mtb cell wall after they have been secreted [210]. Aptamer (Apt) is nonimmunogenic, easier to produce with high purity, and more stable than antibodies. As a result, the Ag85A Apt was used to embellish the niosome surface before its specific therapeutic application. Apt-PEGNio/EEP showed higher anti-Mtb activity than PEGNio/EEP, which might be because the pt-modified niosome surface improves the niosome’s chances of trapping the Ag85A membrane-bound protein. As a result of these findings, Apt has become more popular for identifying the causative agent of the disease. Hence, the pt-functionalized niosome recognizes and binds to the Ag85A membrane-bound proteins of Mtb as the hypothesized mechanism of the developed formulation for anti-TB therapy. Finally, the Apt-PEGNio/EEP penetrates the mycobacterial cell wall and is released in the cytoplasm. EEP’s bioactive chemicals may reduce Mtb viability by interacting with proteins involved in growth and disrupting the cell wall integrity.

## 5. Nanotechnology in the Diagnosis of Tuberculosis

A proper diagnosis of tuberculosis (TB) infection is critical in preventing its spread to another susceptible individual. Patients associated with latent TB infection (LTBI), detected by an immune response to MTB proteins, and are not the focus of TB control activities in endemic regions [211]. People with latent tuberculosis infection (LTBI) are not infectious, but identifying them is crucial, since 10% of these people, especially immunosuppressed ones, may go on to acquire active tuberculosis. Different techniques that are used to diagnose routine TB infections are smear microscopy; MTB culture, detection, and amplification of MTB nucleic acids; and clinical symptoms. However, the tuberculin skin test (TST) and interferon gamma release tests are used to detect LTBI. Three main constraints of the current TB diagnosis are (i) the poor specificity of clinical diagnosis, (ii) lack of high-performance diagnostic tools, and (iii) inability to monitor patients undergoing a 6–9-month treatment period.

Although the tuberculosis diagnostic pipeline is quite advanced, the lack of a simplified, instrument-free test continues to be a barrier [212]. Therefore, it is highly desirable to have a simple, sensitive, portable, and affordable MTB detection procedure for active and latent LTB infection [213].

The unique physicochemical (inert and nontoxic) and optical properties of gold nanoparticles (AuNPs) make them the most suited nanomaterial for clinical diagnostics, therapies, and other interdisciplinary research. The presence of suitable optical properties in AuNPs when they bind with antibodies, antigens, and other biomolecules enables their use as a perfect diagnostic tool for pathogen detection. Furthermore, even after antigen immobilization, AuNPs do not impair the functional activity [214]. The surface functionalization of gold nanoparticles enhances the antibody–antigen response, enhancing immunoassay signals and, as a result, diagnostic sensitivity [215]. It provides a simple and cost-effective assay that permits several samples to be tested at the same time. The test showed an exceedingly specific and reliable result, even at the low level of mycobacterial DNA. A colorimetric analysis of the intended genes and sequences from MTB DNA samples using AuNP probes (ssDNA, thiol-linked ssDNA, or modified gold nanoparticles) provides a cost-effective detection approach [216]. The use of AuNPs with DNA probes using the *M. tuberculosis* RNA polymerase component in TB diagnosis was initially described by [217]. A colorimetric detection approach was used to detect the AuNP-embedded RNA polymerase of *M. tuberculosis* at 526 nm. If complementary DNA is present, the nanoprobe solution shows pink (absence of DNA probe aggregation); however, in the absence of complementary DNA, the solution becomes purple (due to the aggregation of the nanoprobe at a high NaCl concentration). In contrast to the diagnostic procedures, such as InnoLiPA-Rif-TB, which provided 100 percent concordance, the AuNP-embedded polymerase approach was more accurate [217]. The AuNP-based DNA probe approach was more sensitive than smear microscopy and can be easily observed for detection. The main benefit of this procedure is that the risks of contamination are extremely low (it is performed in a single tube, which reduces contamination), and it is quick (~15 min per sample).

Further, a comparison with the automated liquid culture system (BD BACTEC^™^ MGIT^™^) and semi-nested PCR, the AuNP probe-based approach has higher sensitivity and specificity in the identification of the *M. tuberculosis* complex [218]. The *M. tuberculosis* insertion sequence (IS6110) was employed to enhance the sensitivity of this test [219].

According to the IUPAC, “a biosensor is a biomolecule-integrated device that is capable of giving semi-quantitative or quantitative information by utilizing a biomolecular recognition element in direct contact with a transducer element” [220]. Chemical or biological events in a living environment may be detected, quantified, and transduced in a biosensing module using a specialized biosensing receptor that undergoes a change upon binding the molecules of interest. Subsequently, these changes are transformed into a detectable signal [221,222]. Label-embedded and label-free detection are the two primary categories of technological solutions used in biosensor design. The response of biomolecules binding to the recognition element can be accomplished in label-free modules without using radioactive or fluorescent labels or enzymatic tests [223]. In contrast to label-embedded systems, label-free biosensors require large amounts of expensive chemicals and labels, making them a less cost-effective TB diagnostic approach. Due to their direct and real-time multiplexed detection capabilities, label-free biosensors are becoming the most acceptable diagnostic tool [224,225].

Localized surface plasmon resonance (LSPR) biosensors have lately received a lot of interest for point-of-care MTB detection. Compared to SPR systems, the LSPR systems offer a simpler and more compact setup. When placed extremely close to the LSPR substrate, nanoparticles may increase the sensitivity of LSPR biosensors and cause large variations in the LSPR peak location. A recent study that used DNA aptamers with AuNP to build LSPR effectively detected 0.1 nM IFN for the detection of LTB infection [226]. In another case, the fusion protein CFP10-ESAT6 was used as an antigen to construct a LSPR-based biosensor to detect *M. tuberculosis* antibodies. The antigen fusion protein CFP10-ESAT6 was immobilized on Au NRs and functionalized Au NRs, then incubated with serums obtained from TB patients. The LSPR characteristics of Au NRs undergo changes as a result of the interaction between the antigen and the target antibody. The study revealed the sensitivity of the biosensor was 79%, with 92% specificity [227].

The most promising option for TB point-of-care detection is the AuNP-based Mtb detection method, which uses a target DNA amplification step and takes approximately three hours from sample collection to characterization. Further, it takes two hours and 30 min for PCR and approximately thirty minutes for the colorimetric assay [228]. The magnetic immunoassay (MPI) developed by Kim et al. has the potential to detect CFP-10 in positive samples in approximately 10 min [229]. Plasmon absorbance dropped and the red hue diminished when CFP-10 was added to the solution. The test had a detection limit of 10 pg mL^−1^ and a detection range of 10 to 10,000 pg mL^−1^. Every lab-scale diagnostic module aims to achieve the high-throughput execution of any research-scale diagnostic technique. The AuNP-based colorimetric method built in paper microplate wells from this viewpoint. It was proven that the method’s sensitivity and specificity could be demonstrated via the use of a camera on a typical smartphone device or other accessible detectors [230]. A lab-on-a-chip microfluidic device might be equipped with gold nanoprobes to minimize the amount of reagents and samples while also lowering the overall cost. Disposable, affordable microfluidic chips with a concentration of 30 ng L1 of target DNA were constructed for the diagnostic purpose. Only three microliters of the DNA solution were used for this purpose, i.e., 20 times less solution volume was needed for the bulk version of the AuNP-based biosensor [231]. Indeed, microfluidic channels have recently attracted considerable attention in POC diagnostics. Since they are designed to contain all detection components in micro-scale channels, microfluidics provides a user-friendly and cost-effective detection tool [232]. Colorimetric techniques are extremely distinct and receptive screening tools for the identification of mycobacteria among label-free approaches for TB detection [233].

A new method for detecting gene mutations has been developed utilizing magnetic beads. MTB rpoB gene mutations may be detected using magnetic nanobeads tagged streptavidin and biotin-labeled. Using an array of 11 padlock probes (PLPs) aimed at the 23S ITS region, the MTB complex and wild type were both detected, while the RRDR-rpoB gene’s common mutations were identified using the remaining nine probes. AC susceptometry was used to identify the signals, which were identified using the Brownian relaxation technique [234]. Additionally, SPIONs increased the MRI system’s sensitivity and specificity. To create conjugates, anti-MTB surface antibodies were used to activate SPIONs, which were then used to study host–pathogen interactions at the molecular level. Incubation with mycobacterium and imaging were previously performed on this conjugated complex. For extrapulmonary TB, such as central nervous system and abdominal TB, particular target identification would be shown by the lowered signal intensity [235,236].

With this progress in magnetic nanoparticle technology, magnetic barcode assay applications replicating the functional characteristics of quantum dots have been demonstrated. Probes for better TB detection would be specific complementary DNA sequences of MTB [237]. DNA extraction and PCR amplification are no longer necessary thanks to this strategy. NMR (nuclear magnetic resonance) methods were used to identify the conjugate after the DNA was trapped by the MNP probes [238].

## 6. Nanotechnology in the Prevention of Tuberculosis

In the development of nanomaterial vaccines against HIV, malaria, and tuberculosis, the distribution of the vaccines to immune system cells and tissues is an important consideration. In contrast, a nanomaterial vaccine must be able to interact with a wide range of cells, unlike conventional drug delivery applications that may only target a single cell type. APCs, B cells, neutrophils, macrophages, and T cells of different kinds may all be present in the body after the vaccine-associated antigens have been digested. These interactions take place over long periods of time in a variety of tissues, and after extensive processing, make it difficult to rationally design the trafficking of a vaccine nanoparticle. An array of properties may be controlled in vivo, including the nanomaterial’s dimensions and form, as well as its physical orientation, antigen copy number on or inside the nanomaterial, and complement activation. Each of these variables has a significant effect on the quality and potency of the immune responses induced by the vaccination as it travels to lymphoid tissues.

There are two immunodominant antigens of *M. tuberculosis* in the vaccine produced by Yu and colleagues (Ag85A and ESAT-6) and IL-21, which is coated in ion (III) oxide (Fe_2_O_3_) [239]. Immune and cell responses in mice were enhanced by the coated vaccination, as compared to the naked plasmid DNA. It was also shown that animals inoculated with the NP-coated vaccine had a lower lung bacterial load than animals immunized with the naked plasmid vaccine or BCG.

Vaccines use cationic liposomes, particularly DNA delivery vesicles, to improve the immune identification of inert or less immunogenic protein components. Liposome-forming lipids that have been extensively studied include 1, 2-dioleoyl-3-trimethylammonium propane (DOTAP), dimethyldioctadecylammonium (DDA), and 3-[N-(N′, N′-dimethylaminoethane) carbomyl] cholesterol (DCChol). A variety of immunomodulators, including trehalose-6, 6′-dibehenate (TDB), have been tested in conjunction with DDA. Antiviral immune responses against influenza, chlamydia, erythrocytic-stage malaria, and tuberculosis (TB) have been successfully induced by CAF01, which was developed as a combination of DDA and TDB [240].

It is improbable, however, that any new generation of adjuvants will be based only on a single component because of the intricacy of the immune response. Combinations of various immunotherapeutic drugs that are plausible to produce complex and adequate immune responses are viewed as the preferred adjuvant formulations in the future [241]. Polar lipid fractions, commonly known as archaeosomes, are an example of this development. In vitro studies have shown the ability of the archaeosome adjuvant system to induce both a humoral and a cell-mediated immune response [242].

Liposomes have also been used to aid in the retention of medicinal substances in the lungs. Using APC-mediated plasmid uptake as proof, researchers have shown that liposomes improve the efficacy of DNA vaccinations [243]. Monkeys’ lungs were successfully implanted with lipid/DNA complexes during gene therapy [244]. Liposomal material is transported into cells primarily by membrane fusion and endocytosis, two newly identified mechanisms [245]. As a result of these findings, it seems that genetic immunization, in combination with innovative delivery vehicles, may be used to overcome many of the issues associated with traditional vaccine administration and the immune response.

An adjuvant system that simultaneously induces a CMI and antibody response with higher levels of IgG2 antibodies has been developed by CAF01. Post-immunization with the subunit TB vaccine, these antibody responses lasted for over a year [246]. To compare CD4 T cells with CD8 T cells, CD4 T cells are more polyfunctional and highly antigen-specific, get recruited at a greater rate, and undergo higher proliferation after MTB infection compared to CD8 T cells. When exposed to a live infection, CD4 and CD8 T cells respond considerably differently, which supports the theory that the early phases of Mtb infection are dominated by the CD4 T cells’ immune response [247]. Aside from its immune-stimulating properties (thanks to its synergistic impact), TDB also has a significant stabilizing effect on DDA liposome formulation, allowing it to retain its particle size distribution for almost 1.5 years at 4 °C. The TB vaccine candidate Ag85BESAT6 was to be tested with this formulation in clinical trials at the same time [248]. Combining BCG paste with DDA–D (+) TDB (CAF01) adjuvant produces better and safer formulations that promote much higher antituberculosis protective immunity compared to BCG alone. For the adjuvant BCG vaccine formulations to be confirmed as safer and more immunogenic, it is critical that new, low-cost, safe, and immunogenic TB immunization techniques be developed [249].

An autophagy-inducing plasmid was included in a DNA TB vaccination using chitosan nanoparticles in one study. This technique improved the immunological response. A plasmid DNA-loaded chitosan nanoparticle (NP) was produced and tested for immunity against tuberculosis in mice [250]. Autophagy-inducing elements may be added to DNA-loaded chitosan vaccines to enhance powerful immune responses, according to the research [251]. Recently, it was shown that administering a gene vaccine in chitosan formulation via the mouth increased specific SIgA levels and mucosal IFN-(+) T-cell responses, both of which were positively associated with immunity against TB in the long term [252].

When two techniques are combined, the antigen-encoding pDNA may be administered successfully through the nasal route to induce an efficient cellular and humoral immune response. The pDNA is efficiently delivered to NALT (nasal-associated lymphoid tissue) and to the nucleus of APCs through these two methods. A single layer of follicle-associated epithelium (FAE), a kind of epithelial cell sheet, is thought to cover NALT and play a critical role in the mucosal immune response [253]. Using ligands for the FAE to transport vaccine antigens to the mucosal surface is a powerful method for vaccination efficacy. The tight junction transmembrane protein claudin-4 is necessary for the FAE to develop a high level of transepithelial electrical resistance. Nasal vaccinations may benefit from claudin-4-targeting, according to these data. For DNA-embedded antigen delivery to nasal mucosal immune inducing sites, *Clostridium perfringens* enterotoxin (CPE) may be used [254].

## 7. Future Perspectives and Conclusions

Only a few medications, such as quinolones and rifampicin, have made significant advances in TB treatment. Several anti-TB medications are being developed, but their main drawbacks include expensive costs, a paucity of research, drug toxicity, and the difficulty of targeting MDR and dormant bacilli. The need for a new and unique medicinal medicine has risen dramatically because of these factors. Nanotechnology integration is one of the promising measures being pursued right now to improve TB immunization and other respiratory diseases [255].

For mono- and multidrug delivery to treat tuberculosis, several nanocarriers and biomaterials have been studied. The nanosystems not only demonstrated their ability to overcome physicochemical constraints but they also improved drug bioavailability in TB-infected animal models. Oral delivery of these anti-TB medications has been extensively studied and used; in addition, considerable efforts have been made to avoid parenteral administration [256]. Nonetheless, given that pulmonary TB is the most frequent form of TB, inhaled TB therapy using tailored drug delivery systems represents an intriguing alternative to the oral route. Due to its advantages, pulmonary medication delivery has received a lot of attention in recent years. When compared to oral medication, inhalation provides a direct drug delivery to the site of action with fewer systemic side effects. The medications are transported directly to the lungs, resulting in a higher drug concentration at the infection’s primary site. The medications are rapidly absorbed due to the lung’s huge surface area, considerable blood flow, and exceptionally thin alveolar-capillary membrane. This allows for the use of lower medication doses than would otherwise be necessary.

Nanotechnology provides a viable option for producing a respiratory adjuvant to oral TB medication [257]. Their nano-size can help to improve the apparent solubility of medications and achieve a longer-lasting release profile. Nanocarriers can also promote drug penetration via membranes and cellular absorption [257]. The aerodynamic behavior of the nanoparticles is also an important factor to consider. It is a critical goal to evaluate deep lung accumulation and avoid fast clearance from the respiratory system. In this regard, various biomaterials (e.g., PEG-based copolymers) and chitosan (e.g., chitosan) can facilitate mucus penetration and promote muco-adhesion [258,259]. However, due to the lungs’ lack of buffer capacity, the manufacture of inhalable powders, solutions, or suspensions necessitates the consideration of several parameters, such as particle agglomeration; precipitation; and the inclusion of antioxidants, preservatives, buffers, and tonicity agents. Furthermore, there are only a few compounds for pulmonary administration on the existing list of pharmaceutical additives [260]. Many other researchers have also reviewed the nano-TB options for treatment, providing a deep overview of the knowledge and technologies available [261,262,263,264,265]. 

The potential for nanotechnology to enhance tuberculosis therapy is clear, and further study in this field will facilitate the development of clinical applications. Designing anti-TB medicines requires first establishing the capacity of nanoparticles to regulate biological systems such as mitochondrial function, antioxidant activities, and inflammatory responses. The development of host-directed therapeutics may take advantage of the fact that certain nanoparticle formulations can influence host–cell responses, for example, by increasing the immunometabolism to build effective immune responses against the pathogen. Site-targeted medication administration for efficient bacterial clearance may be made possible by nanoparticles’ capacity to localize into subcellular cellular compartments, such as mitochondria and lysosomes [266].

It should be noted that nano-based systems require a significant amount of lyoprotector, which could increase the product’s cost-related variables. It is also worth noting that the overall amount of biomaterial and lyoprotector inhaled over time should be evaluated for chronic lung toxicity. Despite recent advancements in nanotechnology-based TB therapy, the development of commercially accessible nanoformulations is still in its early stages. Only a small number of studies have demonstrated successful formulations in infected preclinical models of tuberculosis, which is a significant issue because subsequent clinical studies would rely on the preclinical results.

## Figures and Tables

**Figure 1 pharmaceuticals-16-00581-f001:**
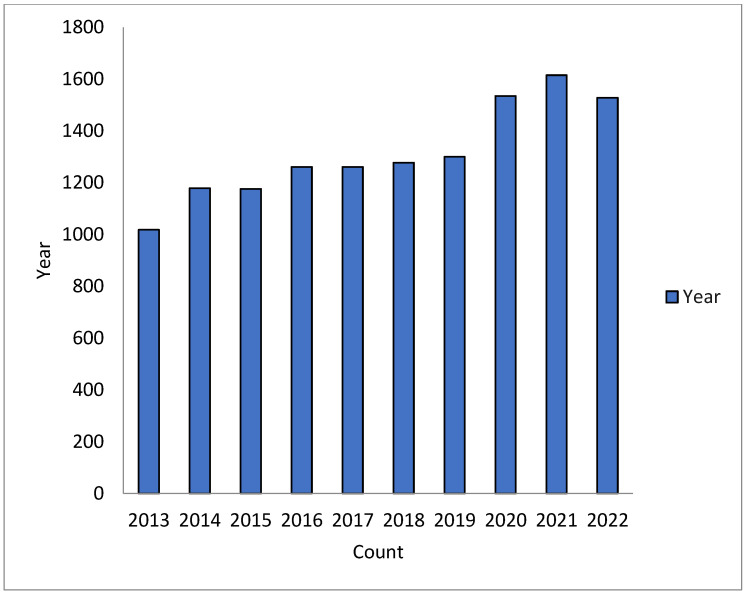
Total number of publications searched using PubMed (as on dated 21 March 2023, term searched “Tuberculosis”).

**Figure 2 pharmaceuticals-16-00581-f002:**
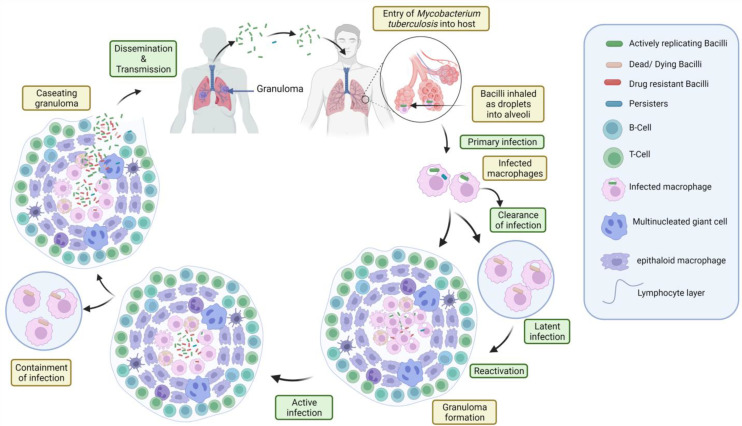
Various stages of TB progression (Created with BioRender.com).

**Figure 3 pharmaceuticals-16-00581-f003:**
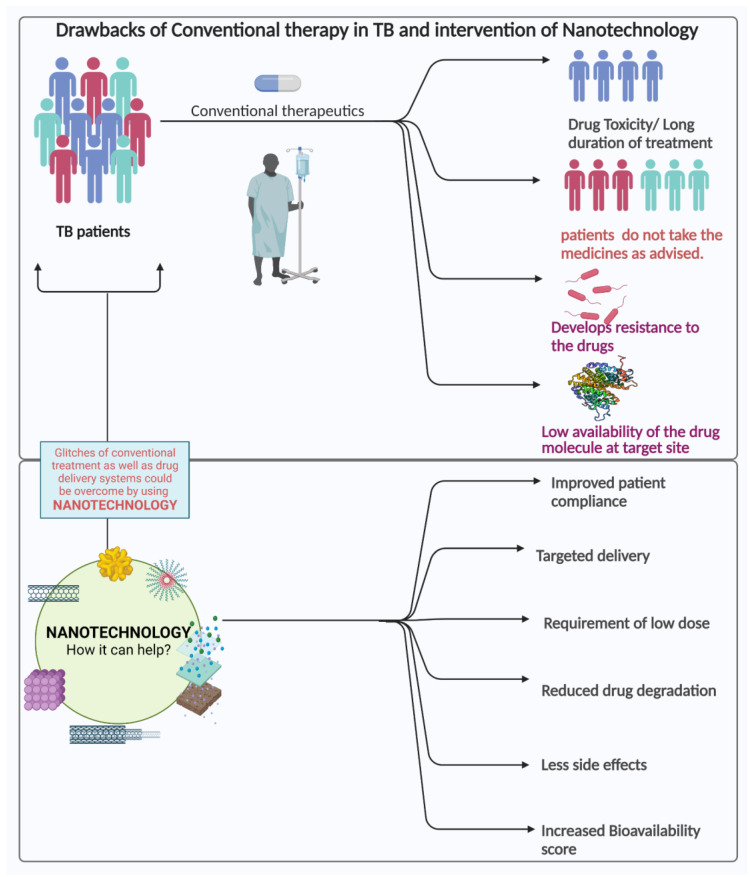
Comparison of traditional vs. nanosystems for the treatment of TB (Created with BioRender.com).

**Figure 4 pharmaceuticals-16-00581-f004:**
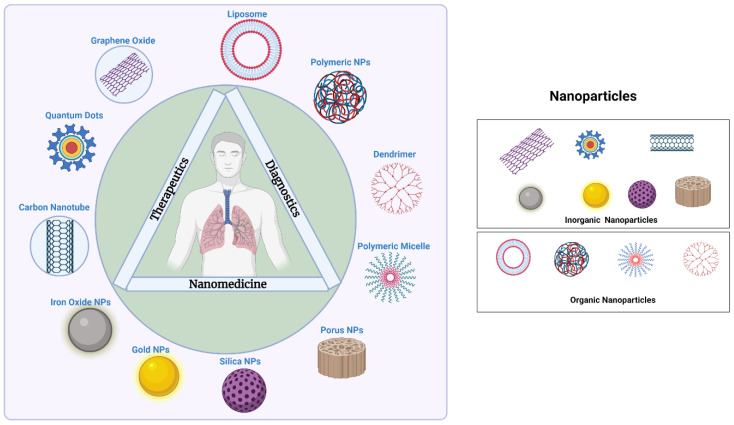
List of the different nanocarriers used in TB diagnosis and treatment (Created with BioRender.com).

**Table 1 pharmaceuticals-16-00581-t001:** Different nanoparticle-based treatments of TB.

Name of Active Nanoparticles	Size and Dimensions	Species of Mycobacterium Targeted	Outcome of Research	Reference
ZnO nanoparticles from *Canthiumdicoccum*	average size = 33 nm; zeta potential 7.3 mV	*M. tuberculosis* (ATCC No-27294)	Anti-TB activity by Alamar Blue Dye test revealed phytofabricated ZnO-NPs inhibited *M. tuberculosis* at 25 μg mL^−1^	[163]
Quaternized chitosan/silver nanocomposites	Spherical 11 to 17.5 nm	*M. tuberculosis* (ATCC 25177)	Inhibition of growth. Disruption of the bacterial cell wall.	[164]
Silver chloride nanoparticles	Spherical 9 to 51 nm	*M. tuberculosis* (H37Ra, and MDR/XDR strains)	Inhibition of growth.	[165]
Isoniazid–Selenium Nanoparticles	Spherical 40 to 45 nm	*M. tuberculosis* strain H37Rv	Isoniazid-linked mannosylated selenium nanoparticles induced autophagy sequestration of Mtb, evolving into lysosome-associated Autophagosomal Mtb degradation linked to ROS-mitochondrial and PI3K/Akt/mTOR signaling pathways.	[166]
Mannosylated gelatin nanoparticles and licorice	237.2 ± 5.11 to 289.6 ± 3.97 nm	*M. tuberculosis* strain H37Rv	Showed statistically significant reduction in bacterial counts in lungs and spleen of *Mycobacterium tuberculosis* H37Rv infected mice as compared to untreated animals.	[167]
Silica nanoparticles and isoniazid	50 nm		Role in increasing the activation of immune cells through the attachment of particles, which will supplement the drug efficacy of INH.	[168]
Magnetic iron oxide nanoparticles (MIONs) cross-linked polyethylene glycol hybrid chitosan and Rifampicin	70.20 ± 3.50 nm		Magnetic gel beads show higher nano drug releasing efficacy at acidic medium (pH = 5.0) with a maximum efficiency of 71.00 ± 0.87%. This efficacy may also be tuned by altering the external magnetic field and the weight percentage (wt%) of PEG	[169]
Amphiphilic chitosan–grafted- (cetyl alcohol-maleic anhydride-pyrazinamide), silver nanoparticles and rifampicin	141.4 ± 1.61 nm and zeta sizer −8.44		pH-dependent drug release, and this is of great potential for drug targets in a lysozyme environment.	[170]
Alginate modified-PLGA nanoparticles entrapping amikacinand moxifloxacin	640 ± 32 nm	*M. tuberculosis* H37Ra	Dual-loaded formulation revealed an enhanced inhibition of viable bacterial count compared to single drug-loaded nanoparticle formulations and untreated cells.	[171]
Ag, ZnO, and Ag-ZnO NPs	5.4 ± 2.6 nm (Ag NPs) and 9.3 ± 3.9 nm (ZnO NPs)	MDR, XDR, and H37Rv (ATCC 27294) strains of *M. tuberculosis*	One microgram per milliliter of Ag and ZnO NPs can inhibit the growth of the XDR strains of *M. tuberculosis.* Moreover, 1–64 μg/mL of various dilutions of Ag-ZnO NPs can inhibit the MDR and H37Rv strains of *M. tuberculosis*.	[107]
Rifampicin-loaded solid lipid nanoparticles	456 ± 11 nm		In vitro GI stability studies (at pH 1.2, pH 4.5, pH 6.8, and pH 7.4) revealed that the developed system could withstand various gastrointestinal tract media	[140]
Titanium dioxide (TiO_2_) nanoparticles		*M. tuberculosis*, *M. bovis* and *Mycobacterium* sp.	The metabolic activity of mycobacteria was decreased up to 3–4-fold, with an increase in the concentration of the TiO_2_ nanoparticles hence affecting the biofilm formation.	[172]
INH (conjugated), clofazimine (CFZ), coumarin-6, 1,8-Octanediol-dimethyl 2-oxoglutarate copolymer NPs	284 ± 11 nm	*M. marinum* strain M carrying pTEC27	Rapid, high-level accumulation in monocytes and neutrophils, and less efficient uptake by B and T cells in human PBMC; colocalization with Mtb H37Rv in phagosomes of dTHP-1; in a *M. marinum*-infected zebra fish model, NPs were taken up by macrophages; NPs had better activity in reducing bacterial burden and granuloma number.	[173]

## Data Availability

The data and information related to this article can be obtained by appropriate official procedure from Dr. Yugal Kishore Mohanta.

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
