# Peer review of "An Insight into Advances in Developing Nanotechnology Based Therapeutics, Drug Delivery, Diagnostics and Vaccines: Multidimensional Applications in Tuberculosis Disease Management"

_pharmaceuticals, 2023, doi:10.3390/ph16040581_

Round 1
Reviewer 1 Report
This manuscript investigated the role of nanotechnology based therapeutics in tuberculosis management. This article claims that using of novel nanoparticles could be a suitable for biomedical applications. Therefore, I suggest a minor correction and require a detailed clarification. Correction to be addressed by the authors as follows: The abstract is not well organized, where the sentences are incomplete and no continuity is there. It would be feasible, if include the significance of the current study in the abstract. A brief description of how the authors selected information from the literature in the databases, as well as doses.
Authors should justify and expand the information on the biomedical application of green nanoparticles in tuberculosis , highlighting the main contribution in in vitro fields. Authors should specify the main experimental conditions used on the evidences from the literature. Where they briefly describe the most important data reported in the literature in a homogeneous manner and sequence reinforcing the relevance of green nanomaterials and plant based nanoparticles as medicinal alternative.
The most significant mechanism of action of this nanoparticles should be described and noticed more emphatically. Authors should discuss whether the use of these nanoparticles represents a solid alternative to existing commercial drugs or a source of new drugs.
Please add below studies to your manuscript in discussion section and also please discuss about possible toxicity of proposed nanomaterials.
DOI: 10.3390/nano13020269
DOI: 10.2217/nnm-2020-0441
Conclusions should reaffirm the fundamental contribution of this paper.
Author Response
This manuscript investigated the role of nanotechnology based therapeutics in tuberculosis management. This article claims that using of novel nanoparticles could be a suitable for biomedical applications. Therefore, I suggest a minor correction and require a detailed clarification. Correction to be addressed by the authors as follows: The abstract is not well organized, where the sentences are incomplete and no continuity is there. It would be feasible, if include the significance of the current study in the abstract. A brief description of how the authors selected information from the literature in the databases, as well as doses.
Reply: Thanks to reviewer. We had revised the abstract as said in comments.
Authors should justify and expand the information on the biomedical application of green nanoparticles in tuberculosis , highlighting the main contribution in in vitro fields. Authors should specify the main experimental conditions used on the evidences from the literature. Where they briefly describe the most important data reported in the literature in a homogeneous manner and sequence reinforcing the relevance of green nanomaterials and plant based nanoparticles as medicinal alternative.
Reply: Thanks to reviewer. We had added the role of green nanoparticle in tuberculosis in text and cited the recent most researches in field to provide readers the current knowledge.
The most significant mechanism of action of this nanoparticles should be described and noticed more emphatically. Authors should discuss whether the use of these nanoparticles represents a solid alternative to existing commercial drugs or a source of new drugs.
Reply: Thanks to reviewer. We had added the role of green nanoparticle in tuberculosis in text and cited the recent most researches in field to provide readers the current knowledge.
Please add below studies to your manuscript in discussion section and also please discuss about possible toxicity of proposed nanomaterials.
DOI: 10.3390/nano13020269
DOI: 10.2217/nnm-2020-0441
Conclusions should reaffirm the fundamental contribution of this paper.
Reply: Thanks to reviewer. We had added the mentioned texts in our manuscript.
Reviewer 2 Report
All the Figures look BioRender-made (or some similar software). If so, the license must be presented, and BioRender should be cited. From BioRender site:
Whether you’re publishing in a journal, textbook, or simply in a presentation or departmental website, all users must cite BioRender figures with the credit “Created with BioRender.com.”
You can include this wherever it makes sense, like the figure caption, citations list, or acknowledgments.
- Line 56 - The reference regarding citation 3 is not possible to find the way it is in the reference list.
- Line 67 - Reference 7 - The written address in the references list is correct, but the hyperlink to the document is wrong.
- Line 128 - All the MDR and XDR definitions are old. WHO updated the definitions (https://www.who.int/publications/i/item/9789240018662) and now there is a "Pre-XDR" category as well. This change has to be corrected all over the Review since the authors cite MDR and XDR types of TB in several points of the text.
- Line 341 - there is a lonely ".
- Line 384 - The authors say "Rifamycin". The intention was to talk about Rifampicin or about the Rifamycin class of drugs (Rifampicin, Rifabutin, Rifapentin....). I think that Rifampicin (or Rifampin) is the intention. If so, please change it.
Line 405 - A clear connection between all that was said and the next topic (the topic about nanotech) has to be made. Figure 2 states clearly that the solution for the problems presented in this paragraph is at least partially in nanoscience. Insert this information in this paragraph.
Line 418 - This Review can be more complete. A definition of nanoparticles based on the literature can be inserted here. Characteristics of inorganic and organic nanoparticles can be explained, pointing out their pros and cons to the subject.
- Lines 421, 423 and 425 - Affirmations are made. Citations that support these affirmations are needed.
- The phrase in line 444 is hard to understand. Yet, in line 446, it says that the particle has 1-5um. Is it the PNAP or the nanoparticle that before was said that it has <100nm?
- Line 448 - Here and in other parts of the text, physio chemical is written. Are the authors talking about physiological and chemical parameters or physicochemical parameters?
- Line 583 - Correct Qurrat-ul-Ain et al. citation
- Line 585 - Correct the measure unity
- Line 587 - Here and in several other parts of the text, there is a mistake in the space between words, before and after citations and dots. All text must be improved regarding this aspect
- Table 1 - change to landscape orientation, let it on a separate page
- Line 894 - Correct the name of the kit (BACTEC MGIT. The "TM" superscript is wrongly linked to the name of the brand)
All over the text and references - check for the italics on M. tuberculosis.
Author Response
All the Figures look BioRender-made (or some similar software). If so, the license must be presented, and BioRender should be cited. From BioRender site:
Whether you’re publishing in a journal, textbook, or simply in a presentation or departmental website, all users must cite BioRender figures with the credit “Created with BioRender.com.”
You can include this wherever it makes sense, like the figure caption, citations list, or acknowledgments.
Reply: Thanks to reviewer. We had added citation to Biorender to all figures prepared using biorender.
- Line 56 - The reference regarding citation 3 is not possible to find the way it is in the reference list.
Reply: Thanks to reviewer. We had corrected the reference 3.
- Line 67 - Reference 7 - The written address in the references list is correct, but the hyperlink to the document is wrong.
Reply: Thanks to reviewer. We had corrected the reference 7.
- Line 128 - All the MDR and XDR definitions are old. WHO updated the definitions (https://www.who.int/publications/i/item/9789240018662) and now there is a "Pre-XDR" category as well. This change has to be corrected all over the Review since the authors cite MDR and XDR types of TB in several points of the text.
Reply: Thanks to reviewer. We had corrected and updated the definition as per suggested citation.
- Line 341 - there is a lonely ".
Reply: Thanks to reviewer. We had corrected the sentence.
- Line 384 - The authors say "Rifamycin". The intention was to talk about Rifampicin or about the Rifamycin class of drugs (Rifampicin, Rifabutin, Rifapentin....). I think that Rifampicin (or Rifampin) is the intention. If so, please change it.
Reply: Thanks to reviewer. We had corrected the sentence.
Line 418 - This Review can be more complete. A definition of nanoparticles based on the literature can be inserted here. Characteristics of inorganic and organic nanoparticles can be explained, pointing out their pros and cons to the subject.
Reply: Thanks to reviewer. We had added the text and updated with required information.
- Lines 421, 423 and 425 - Affirmations are made. Citations that support these affirmations are needed.
Reply: Thanks to reviewer. We had corrected the text.
- The phrase in line 444 is hard to understand. Yet, in line 446, it says that the particle has 1-5um. Is it the PNAP or the nanoparticle that before was said that it has <100nm?
Reply: Thanks to reviewer. We had corrected the text.
- Line 448 - Here and in other parts of the text, physio chemical is written. Are the authors talking about physiological and chemical parameters or physicochemical parameters?
Reply: Thanks to reviewer. We had corrected the text.
- Line 583 - Correct Qurrat-ul-Ain et al. citation
Reply: Thanks to reviewer. We had corrected the text.
- Line 585 - Correct the measure unity
Reply: Thanks to reviewer. We had corrected the text.
- Line 587 - Here and in several other parts of the text, there is a mistake in the space between words, before and after citations and dots. All text must be improved regarding this aspect
Reply: Thanks to reviewer. We had corrected the text.
- Table 1 - change to landscape orientation, let it on a separate page
Reply: Thanks to reviewer. The change to landscape can be done as per final layout of manuscript.
- Line 894 - Correct the name of the kit (BACTEC MGIT. The "TM" superscript is wrongly linked to the name of the brand)
Reply: Thanks to reviewer. We had corrected the text.
All over the text and references - check for the italics on M. tuberculosis.
Reply: Thanks to reviewer. We had corrected the text.
Reviewer 3 Report
The authors are advised to include more recent references or resarch findings on the topic (from recent years), and by presenting the total number of publications in the last ten years till date in a bar diagram - in Introduction section.
Please highlight the major drawbacks or limitations of applying nanotechnology tools in the proposed study, for each methods described in the review.
Few important literature to cite:
- https://doi.org/10.1002/9781394167708.ch11
- https://doi.org/10.1111/jcmm.17677
- https://doi.org/10.1002/adtp.202000113
- https://doi.org/10.1186/s12951-022-01307-x
- https://doi.org/10.3390/pharmaceutics15020393
English language needs a major revision. There are multiple grammatical errors, typos, and incorrect forms throughout the manuscript.
Author Response
The authors are advised to include more recent references or resarch findings on the topic (from recent years), and by presenting the total number of publications in the last ten years till date in a bar diagram - in Introduction section.
Reply: Thanks to reviewer. We had added the figure 1.
Please highlight the major drawbacks or limitations of applying nanotechnology tools in the proposed study, for each methods described in the review.
Reply: Thanks to reviewer. We had added the text where as applicable in text.
Few important literature to cite:
- https://doi.org/10.1002/9781394167708.ch11
- https://doi.org/10.1111/jcmm.17677
- https://doi.org/10.1002/adtp.202000113
- https://doi.org/10.1186/s12951-022-01307-x
- https://doi.org/10.3390/pharmaceutics15020393
English language needs a major revision. There are multiple grammatical errors, typos, and incorrect forms throughout the manuscript.
Reply: Thanks to reviewer. We had added the given reference and corrected the grammatical mistakes.